# The energetic and allosteric landscape for KRAS inhibition

Chenchun Weng[1], Andre J. Faure[1], Albert Escobedo[1] & Ben Lehner[1,2,3,4 ✉]

Thousands of proteins have been validated genetically as therapeutic targets for human diseases[1]. However, very few have been successfully targeted, and many are considered 'undruggable'. This is particularly true for proteins that function via protein–protein interactions—direct inhibition of binding interfaces is difficult and requires the identification of allosteric sites. However, most proteins have no known allosteric sites, and a comprehensive allosteric map does not exist for any protein. Here we address this shortcoming by charting multiple global atlases of inhibitory allosteric communication in KRAS. We quantified the effects of more than 26,000 mutations on the folding of KRAS and its binding to six interaction partners. Genetic interactions in double mutants enabled us to perform biophysical measurements at scale, inferring more than 22,000 causal free energy changes. These energy landscapes quantify how mutations tune the binding specificity of a signalling protein and map the inhibitory allosteric sites for an important therapeutic target. Allosteric propagation is particularly effective across the central β-sheet of KRAS, and multiple surface pockets are genetically validated as allosterically active, including a distal pocket in the C-terminal lobe of the protein. Allosteric mutations typically inhibit binding to all tested effectors, but they can also change the binding specificity, revealing the regulatory, evolutionary and therapeutic potential to tune pathway activation. Using the approach described here, it should be possible to rapidly and comprehensively identify allosteric target sites in many proteins.

The GTPase KRAS is somatically mutated in around 10% of all cancers, including about 90% of pancreatic adenocarcinoma, 40% of colorectal adenocarcinoma, 35% of lung adenocarcinoma and 20% of multiple myeloma[2]. KRAS functions as an archetypal molecular switch, cycling between inactive GDP-bound and active GTP-bound states. The altered conformation and activity of KRAS upon GTP binding is an example of allostery, the long-range transmission of information from one site to another in a protein[3]. Many structures of KRAS have been determined, revealing major (but variable) rearrangements in the switch-I and switch-II regions that allow binding to effector proteins in GTP-bound states[4]. KRAS effectors include the RAF proto-oncogene serine/threonine protein kinase (RAF1 (also known as CRAF)), phosphatidylinositol 4,5-bisphosphate 3-kinase catalytic subunit-γ isoform (PIK3CG) and the signalling protein RAL guanine nucleotide dissociation stimulator (RALGDS). Guanine nucleotide exchange factors such as SOS1 catalyse the release of GDP to activate KRAS whereas GTPase-activating proteins (GAPs) catalyse GTP hydrolysis to complete the cycle back to the inactive states. Cancer driver mutations interfere with this cycle, increasing the abundance of active GTP-bound effector-binding states[5,6].

Despite its identification as an oncoprotein more than 40 years ago[7], tens of thousands of scientific publications, and more than 300 published structures of KRAS[4], only recently have inhibitors of KRAS been approved for clinical use, pioneered by sotorasib, a covalent binder of the driver mutation KRAS(G12C)[8–10]. Sotorasib is an allosteric inhibitor that binds outside of the nucleotide and effector binding sites to lock KRAS(G12C) in inactive GDP-bound states, reducing effector binding and clinically validating the efficacy of allosteric KRAS inhibition[8,10]. Similar to many other medically important proteins, the development of therapeutic agents that target KRAS is limited by the lack of information about inhibitory allosteric sites to target. Indeed, a comprehensive map of allosteric sites has not been generated for any oncoprotein or indeed for any disease target protein or any complete protein in any species.

Atlases of allosteric sites have the potential to greatly accelerate drug development, especially for the many human proteins considered undruggable because of the lack of an appropriate active site or because they function via difficult-to-inhibit protein–protein interaction interfaces. In addition, among other benefits, allosteric drugs often have higher specificity than orthosteric drugs that target conserved active sites[11,12].

## KRAS biophysics at scale

To comprehensively map inhibitory allosteric communication in KRAS, we applied a multidimensional deep mutational scanning approach[13]. We used two rounds of nicking mutagenesis[14] to construct three libraries of KRAS variants in which every possible single amino acid

[1]Centre for Genomic Regulation (CRG), The Barcelona Institute of Science and Technology, Barcelona, Spain. [2]University Pompeu Fabra (UPF), Barcelona, Spain. [3]Institució Catalana de Recerca i Estudis Avançats (ICREA), Barcelona, Spain. [4]Wellcome Sanger Institute, Wellcome Genome Campus, Hinxton, UK. ✉e-mail: bl11@sanger.ac.uk

substitution is present not only in the wild-type KRAS (4B isoform, amino acids 1–188) but also in KRAS variants with a range of reduced activities (median of ten genetic backgrounds for each single mutant; Fig. 1a–d). Quantifying the effects of the same mutations in different genetic backgrounds (here double amino acid substitutions) provides sufficient data to resolve biophysical ambiguities[15] and infer the causal biophysical effects of each mutation (see below). In total, the library consists of more than 26,500 variants of KRAS, including more than 3,200 single amino acid substitutions and more than 23,300 double amino acid substitutions.

We first quantified the binding of these KRAS variants to the RAS-binding domain (RBD) of the oncoprotein effector RAF1. Binding was quantified using a protein-fragment complementation assay[13,16,17] (BindingPCA). Binding fitness was highly correlated among three independent replicate selections (Pearson's $r > 0.9$; Extended Data Fig. 1a) to previous data that used a different binding assay in a different cellular context[18,19] (Pearson's $r = 0.82$; Extended Data Fig. 1c) and to individual growth measurements (Pearson's $r = 0.94$; Extended Data Fig. 1d).

As expected, mutations in the RAF1-binding interface strongly inhibit binding, as do variants in the nucleotide-binding pocket (Fig. 1e,i). However, 2,019 out of 3,231 single amino acid substitutions reduce binding to RAF1 (false discovery rate (FDR) = 0.05, two-sided $z$-test), including many outside of the interface and in the hydrophobic core of the protein (Extended Data Fig. 1e). This strongly suggests that many changes in binding to RAF1 are caused by changes in the abundance of folded KRAS and not by altered binding affinity[13,20].

## From phenotypes to free energy changes

To disentangle the effects of mutations on KRAS folding and binding, we used a second selection assay, AbundancePCA[13,21], to quantify the cellular abundance of the KRAS variants. We refer to this combined approach of BindingPCA and AbundancePCA as 'doubledeepPCA'[13] (ddPCA). Plotting the RAF1 binding of each variant against its cellular abundance shows that many changes in binding can indeed be explained by reduced KRAS abundance (Fig. 1j). However, inspection of Fig. 1j also reveals that a substantial number of variants have effects on binding that are much larger than can be accounted for by their reduced abundance, including many variants in the binding interface (red dots in Fig. 1j).

Protein folding and binding relate to changes in the free energies of folding ($\Delta G_f$) and binding ($\Delta G_b$) by nonlinear functions derived from the Boltzmann distribution[13,20] (Fig. 1b). Typically, many different combinations of folding and binding energy changes could underlie a measured change in binding due to a mutation. ddPCA is an efficient experimental design to generate sufficient data to infer en masse the causal biophysical effects of mutations. There are three key principles of the approach. First, mutational effects are quantified for multiple phenotypes—here the binding of KRAS to RAF1 and the abundance of KRAS in the absence of RAF1. Second, mutational effects are not only quantified in wild-type proteins but also in genetic backgrounds with altered folding and/or binding energies—here our libraries contain a median of ten double mutants for each single amino acid substitution in KRAS. Third, the data are used to fit a thermodynamic model in which free energy changes due to mutations combine additively in energy space (but not additively for the measured molecular phenotypes; Methods).

We biased the choice of genetic backgrounds in our KRAS library to mutations with weak detrimental effects and used MoCHI, a substantially improved flexible package to fit user-defined mechanistic models to deep mutational scanning data using neural networks[22], to fit a three-state (unfolded KRAS, folded KRAS and bound KRAS) thermodynamic model to the data (Fig. 1c, Extended Data Fig. 1f–k and Methods). The fitted model predicts the double amino acid mutant data held out during training very well (abundance median $R^2 = 0.74$, binding median $R^2 = 0.91$; Extended Data Fig. 1f,g,i,j) strongly supporting the assumption that most free energy changes combine additively in doubles and these inferred free energy changes are highly correlated with in vitro measurements (Pearson's $r = 0.95$; Fig. 1k). Evaluating model performance on a held out test replicate gave similar results (abundance median $R^2 = 0.54$, binding median $R^2 = 0.87$; Extended Data Fig. 1h,k).

## The RAF1-binding interface

In total, 2,241 out of 3,453 single amino acid substitutions are detrimental to folding and 843 out of 3,301 are detrimental to binding (FDR = 0.05; Fig. 1g,h). Mutations detrimental to folding are enriched in the hydrophobic core of the protein (odds ratio (OR) = 1.92, $P < 10^{-16}$; Fig. 1h,l, two-sided Fisher's exact test; Supplementary Video 1). By contrast, mutations that increase the binding free energy are strongly enriched in the binding interface (OR = 6.02, $P < 10^{-16}$; Figs. 1g and 2a), with the mean absolute binding free energy changes upon mutation at each site identifying the binding interface (Fig. 2b,c and Supplementary Video 2, receiver operating curve area under curve (ROC-AUC) = 0.8 compared with ROC-AUC = 0.65 when using the mean absolute binding fitness).

The interface residues that are most important for RAF1 binding include a mixture of charged (E37 and D38) and hydrophobic (I36 and Y40) residues. D38 cannot be changed to any other amino acid without detrimental effects on binding affinity, revealing a requirement for both negative charge and a particular side chain length at this site (Fig. 2d,e). By contrast, E37 can be replaced by D (shortening the side chain but retaining the negative charge) and also by Y, F or H, suggesting that the salt bridge to RAF1 R67 can be replaced by an alternative interaction involving an aromatic side chain. Y40 can only be replaced by F, revealing the importance of the aromatic side chain which makes a cation–π interaction with RAF1 R89. I36 makes two hydrophobic contacts with RAF1, and whereas polar mutations at this position are detrimental, multiple hydrophobic substitutions are tolerated. Mutations at other residues that contact RAF1 are much better tolerated, indicating that these sites are less important for binding. For example, mutations at D33 tend to be mildly detrimental, with only charge-reversing mutations to R and K and mutation to P strongly inhibiting binding. Similarly, charge-reversing mutations and mutation to P are also most detrimental at R41, whereas mutations at the other two charged sites (E31 and E3) at the edge of the interface generally have little effect on the binding free energy.

## Allosteric landscape for RAF1 binding

We next considered mutations outside of the binding interface. In total, there are 361 distal mutations in 74 residues that increase the binding free energy to a greater extent than the average effect of mutations in the RAF1-binding interface ($\Delta\Delta G_b$ greater than the weighted mean absolute binding free energy change of substitutions in binding interface residues, FDR = 0.05; Fig. 3a). Allosteric mutations defined in this manner are highly enriched in the physiological allosteric site of KRAS, the nucleotide-binding pocket (157 mutations in 13 residues, OR = 7.68, $P < 10^{-16}$, two-sided Fisher's exact test).

## Enhanced allosteric communication

We first focused on residues in which many different mutations have strong allosteric effects. Defining major allosteric sites as residues where the mean absolute change in binding free energy upon mutation is equal to or greater than that in binding interface residues identifies a total of 18 sites (Fig. 3b,c). Ten of these major allosteric sites are located in the physiological allosteric site—the nucleotide-binding pocket (Fig. 3b,c). The additional eight major allosteric sites are residues V7, G10, D54, T58, A59, P110, F141 and I163 (Fig. 3b). Three of these residues

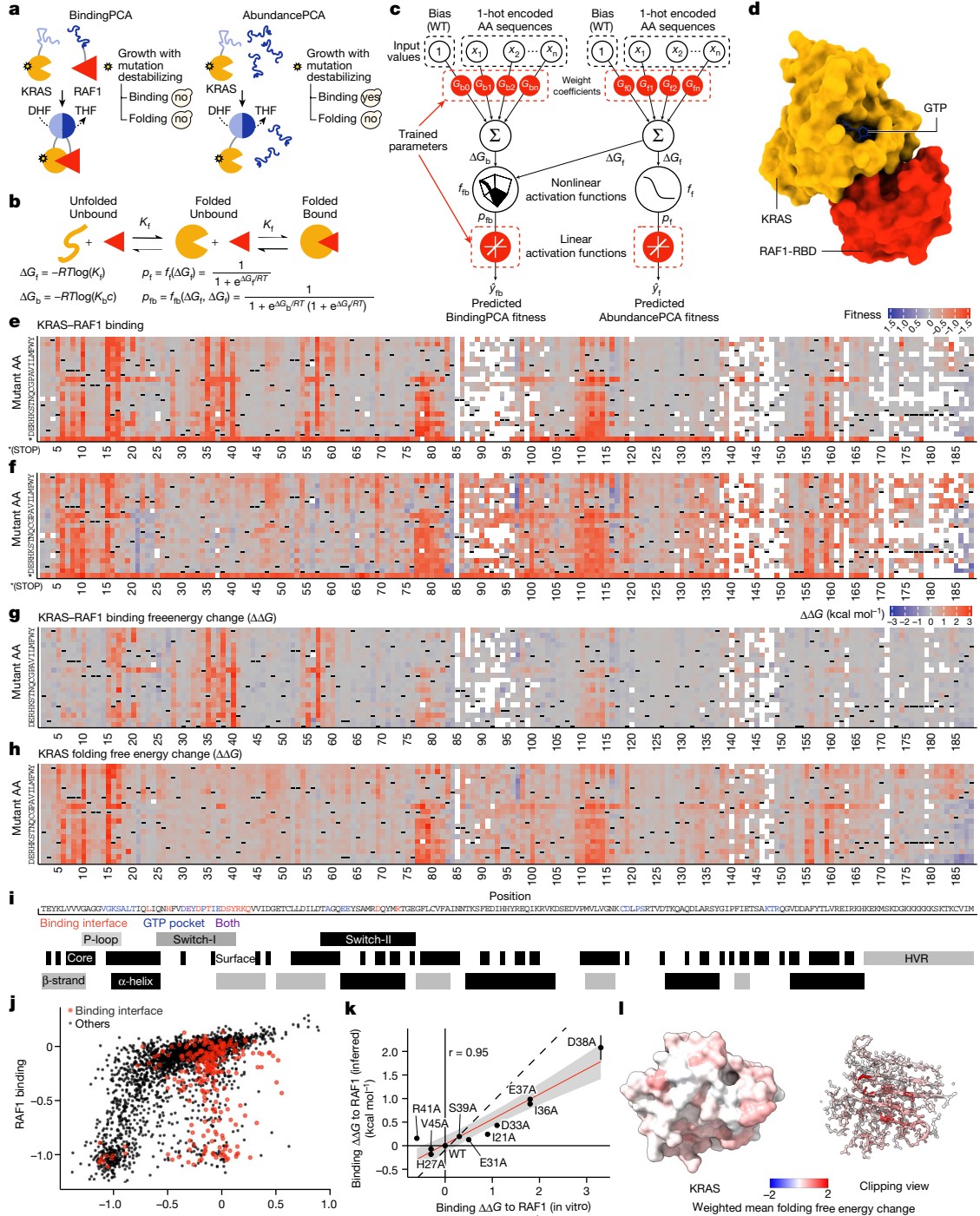

**Fig. 1 | Mapping the energetic landscape of KRAS folding and binding to RAF1. a**, Overview of ddPCA selections. Yes, yeast growth; no, yeast growth defect; DHF, dihydrofolate; THF, tetrahydrofolate. **b**, Three-state equilibrium and corresponding thermodynamic model. $\Delta G_f$, Gibbs free energy of folding; $\Delta G_b$, Gibbs free energy of binding; $K_f$, folding equilibrium constant; $K_b$, binding equilibrium constant; $c$, binding partner concentration; $p_f$, fraction folded; $p_{fb}$, fraction folded and bound; $f_f$, nonlinear function of $\Delta G_f$; $f_{fb}$, nonlinear function of $\Delta G_f$ and $\Delta G_b$; $R$, gas constant; $T$, temperature in Kelvin. **c**, Neural network architecture used to fit thermodynamic models to the ddPCA data (bottom, target and output data), thereby inferring the causal changes in free energy of folding and binding associated with single amino acid substitutions (top, input values). AA, amino acid; WT, wild type. **d**, 3D structure of KRAS bound to the RAF1 RBD (RAF1-RBD) (Protein Data Bank (PDB) ID: 6VJJ). **e,f**, Heat maps of fitness effects of single amino acid substitutions for KRAS–RAF1 from BindingPCA (**e**) and AbundancePCA (**f**) assays. White spaces indicate missing values; dashes are wild-type amino acids; asterisk indicates a stop

codon. **g,h**, Heat maps showing inferred changes in free energies of binding (**g**) and folding (**h**). **i**, Sequence and annotation of KRAS. Binding interface is defined by RAF1 distance <5 Å; GTP pocket is defined by GTP or $Mg^{2+}$ distance <5 Å; core is defined by relative accessible surface area < 0.25; based on PDB ID: 6VJJ. P-loop, residue numbers 10–17; switch-I: 25–40; switch-II: 58–76; α-helix 1: 15–24; α-helix 2: 67–73; α-helix 3: 87–104; α-helix 4: 127–136; α-helix 5: 148–166; β-strand 1: 3–9; β-strand 2: 38–44; β-strand 3: 51–57; β-strand 4: 77–84; β-strand 5: 109–115; β-strand 6: 139–143. **j**, Scatter plot comparing abundance and binding fitness of single amino acid substitutions. Substitutions in the binding interface are indicated in red. **k**, Comparisons of model-inferred free energy changes to in vitro measurements[40]. Error bars indicate 95% confidence intervals from a Monte Carlo simulation approach ($n = 10$ experiments). Linear regression fit and its 95% confidence interval are shown as a red solid line and a grey shaded area, respectively. Pearson's $r$ is shown. Black dashed line indicates $y = x$. **l**, 3D structure (left) and clipping view (right) of KRAS with residues coloured by the weighted mean folding free energy change.

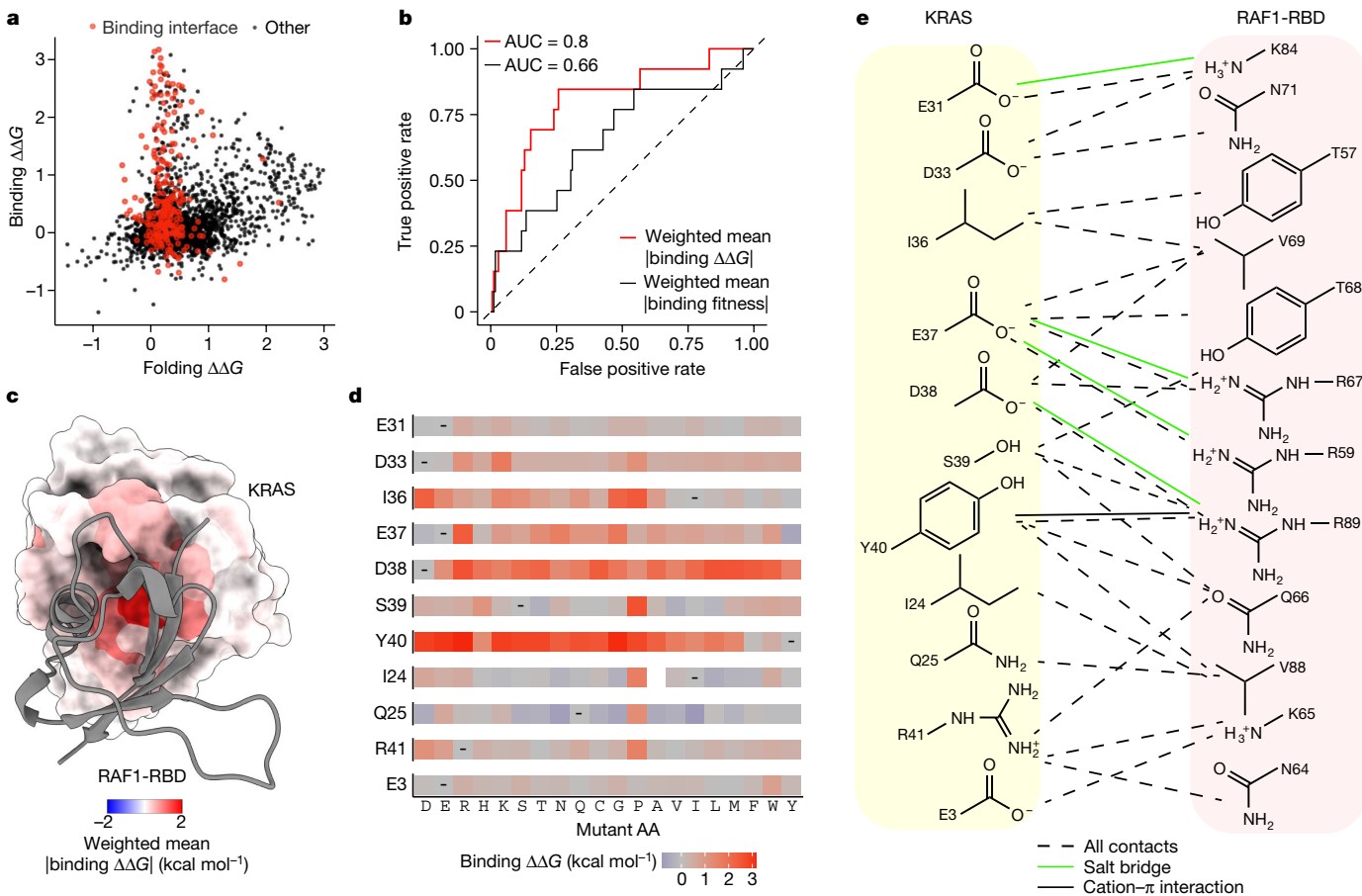

**Fig. 2 | Free energy changes of mutations in the KRAS–RAF1 binding interface. a**, Scatter plot comparing binding and folding free energy changes of single amino acid substitutions. **b**, Receiver operating curves (ROCs) for predicting binding interface residues (RAF1 distance < 5 Å) using weighted mean absolute binding ΔΔG (red) or using weighted mean absolute binding fitness (black). AUC, area under the curve. Dashed line at $y = x$ indicates performance of a random predictor. **c**, 3D structure of KRAS bound to RAF1 in which residue atoms are coloured by the position-wise weighted mean absolute change in the free energy of binding to RAF1. RAF1-RBD is shown in grey ribbon. **d**, Heat maps of binding free energy changes in RAF1-binding interface residues. **e**, Direct contacts between KRAS and RAF1.

are close to the binding interface, with D54 being adjacent to the binding interface and T58 and A59 connecting the binding interface to the nucleotide-binding pocket (Fig. 3c and Supplementary Video 3).

Notably, 5 of the 8 novel major allosteric residues are located in the central (and only) 6-stranded β-sheet of KRAS (Fig. 3b,c, OR = 5.24, $P = 2.8 \times 10^{-2}$, two-sided Fisher's exact test). Within the β-sheet, the binding free energy changes are largest for mutations in residues in the first strand that contacts RAF1 and they progressively decrease in each subsequent strand of the sheet (Fig. 3d,e, Extended Data Fig. 2a–c and Supplementary Video 4). This decay of the strength of allosteric effects across the sheet is consistent with local energetic propagations that underlie allosteric communication. A similar, yet weaker, distance-dependent decay is observed for residues outside of the β-sheet (Extended Data Fig. 2c). Propagation appears more efficient across the sheet than along the backbone within a strand, with residues in the first strand that do not contact RAF1 being depleted for allosteric mutations (Fig. 3a and Extended Data Fig. 2b, OR = 0.16, $P = 10^{-3}$, two-sided Fisher's exact test). Allosteric communication therefore seems to be particularly effective across the central β-sheet of KRAS.

## KRAS has four active surface pockets

We next considered the effects of mutations in the surface residues of KRAS, focusing on the four previously described structural pockets in addition to the nucleotide-binding pocket[23] (Fig. 3f and Supplementary Video 5).

Pocket 1 (also called the switch-I/II pocket) is located behind switch-II between the central β-sheet and α-helix 2 and is the binding site for multiple inhibitors that are effective in pre-clinical models[24,25]. Many mutations in pocket 1 allosterically inhibit RAF1 binding (57 mutations in 10 residues, FDR = 0.05; Fig. 3f and Extended Data Fig. 2d), consistent with the demonstrated ability of pocket 1 engagement to inhibit effector binding.

Pocket 2 (also called the switch-II pocket) is located between switch-II and α-helix 3 and is the binding site of sotorasib and other clinically approved allosteric inhibitors of KRAS(G12C)[26]. Seventy-one mutations in nine residues that contact sotorasib allosterically inhibit RAF1 binding (Fig. 3g and Extended Data Fig. 2e). Thus, mutations and small molecules binding to pocket 1 and pocket 2 can allosterically inhibit KRAS activity.

Pocket 3 of KRAS is located in the C-terminal lobe of the protein and is the most distant pocket from the RAF1-binding interface (Fig. 3f,g). The effects of pocket 3 engagement are not well described[23] and pocket 3 has received little attention for therapeutic development[24]. However, our data reveal that pocket 3 is allosterically active, with 20 mutations in 6 residues in pocket 3 inhibiting binding to RAF1 (Fig. 3g and Extended Data Fig. 2f). The effects of mutations located in pocket 3 were validated in individual growth assays (Pearson's $r = 0.94$; Extended Data Fig. 1d). We also validated the effects on in vitro binding to RAF1 of an allosteric

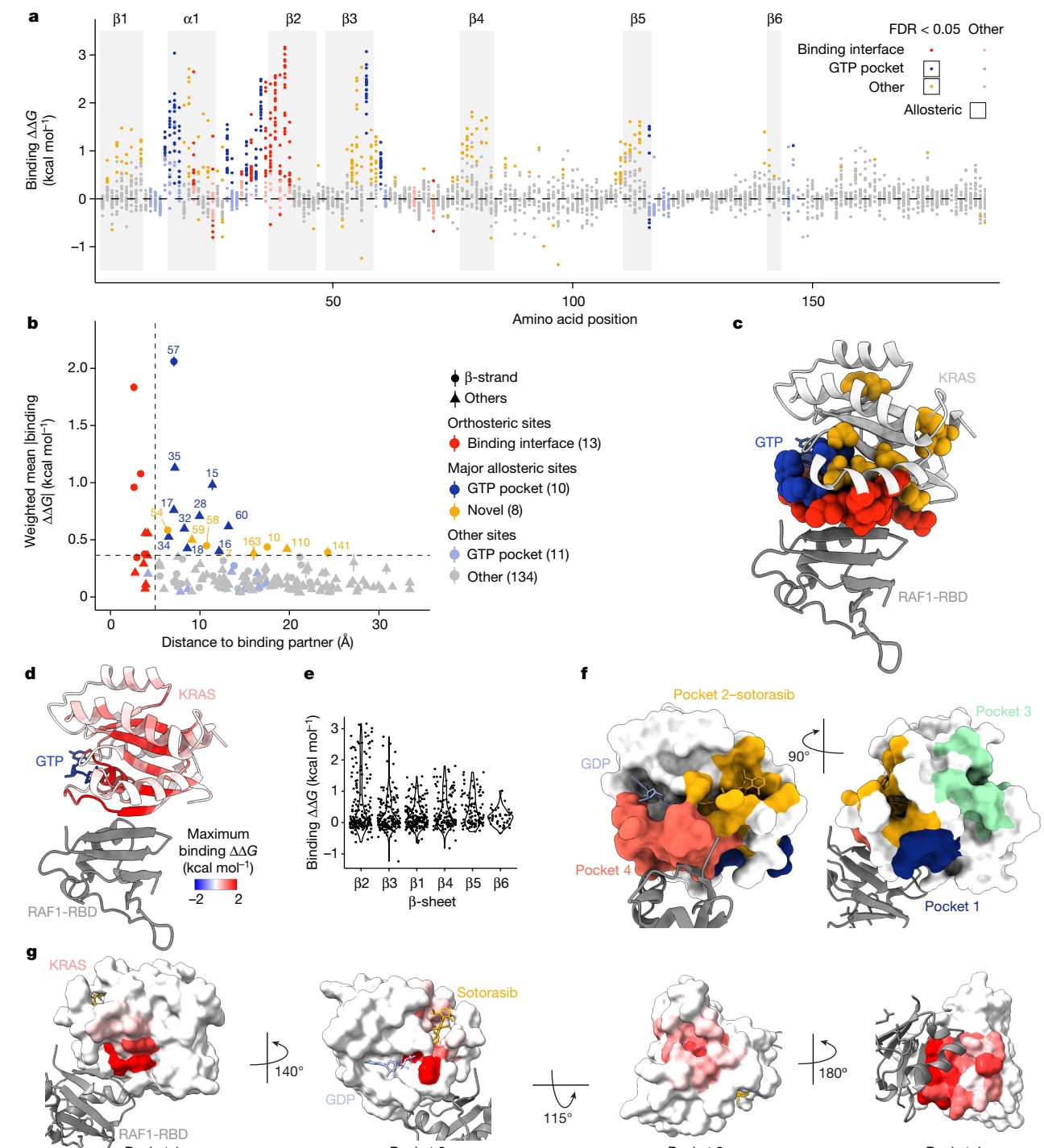

**Fig. 3 | Allosteric regulation of KRAS binding to RAF1. a**, Manhattan plot showing the binding free energy changes of all single amino acid substitutions. Dots are coloured according to residue position and whether the corresponding binding ΔΔ*G* is significantly greater than the weighted mean absolute binding ΔΔ*G* of all mutations in the RAF1-binding interface (two-sided *z*-test, FDR = 0.05). **b**, Relationship between the position-wise average absolute change in free energy of binding to RAF1 and the minimal side chain heavy atom distance to RAF1. Major allosteric sites are defined as non-binding-interface residues with weighted mean absolute change in free energy of binding higher than the average of binding-interface residue mutations (horizontal dashed line). Error bars indicate 95% confidence intervals (*n* ≥ 10). **c**, 3D structure (PDB ID: 6VJJ) of KRAS bound to RAF1 (grey)

with binding interface and major allosteric site residue atoms of KRAS coloured as in **b**. **d**, Similar to **c**, except KRAS residues are coloured by maximum binding ΔΔ*G*. **e**, Violin plot showing the decay of binding free energy change across successive strands in the β-sheet. β-strands are ordered by increasing distance to RAF1 in the 3D structure. **f**, 3D structure alignment (PDB IDs: 6OIM and 6VJJ) of KRAS bound to GDP (blue), sotorasib (yellow) and RAF1 (grey) with KRAS surface coloured according to previously described pockets in KRAS. Pocket 2, sotorasib distance < 5 Å; pockets 1, 3 and 4 (ref. 23); pocket 1, residues 5–7, 39, 54–56 and 70–75; pocket 2, residues 61–65 and 90–94; pocket 3, residues 97, 101, 107–111, 136–140 and 161–166; pocket 4, residues 17, 21, 24–40 and 57. **g**, Similar to **f** except KRAS pockets are coloured by maximum binding ΔΔ*G*.

mutation in pocket 3 (P110F), as well as a mutation in an additional newly discovered major allosteric site (A59R) (Extended Data Fig. 2h). Despite its distance from the effector-binding interface, our data show that pocket 3 should be prioritized as a site for the development of KRAS inhibitors.

Finally, pocket 4, which is located immediately behind the flexible effector-binding loop, contains 105 allosteric mutations in 9 residues that do not contact RAF1 (Fig. 3g and Extended Data Fig. 2g). Our data therefore validate all four surface pockets of KRAS as allosterically active, with perturbations in all pockets having large inhibitory effects on RAF1 binding. This is a strong argument for the development of molecules targeting all four pockets as potential KRAS inhibitors.

## Energetic maps for six KRAS interactions

Similar to most oncoproteins, KRAS binds many different proteins as part of its physiological and disease-relevant functions[3]. Many of these interaction partners bind a common surface of KRAS—the effector-binding interface—making KRAS an interesting model of multi-specificity in molecular recognition[3]. To our knowledge, the effects of mutations on binding energies for multiple interaction partners have not been comprehensively profiled for any protein. Moreover, quantifying KRAS binding to multiple interaction partners provides an opportunity to quantify the conservation and specificity of allosteric effects in a signalling hub (Fig. 4a).

We quantified the binding of the more than 26,000 KRAS variants to six interaction partners: the three KRAS effector proteins RAF1, PIK3CG and RALGDS, the guanine nucleotide exchange factor SOS1, and two DARPins, K27 and K55 (synthetic antibody-like molecules selected to bind GDP-bound KRAS and GTP-bound KRAS, respectively). The structures of all six complexes have been determined[27–31].

The data for all six binding selections were highly reproducible (Extended Data Figs. 1a and 3a), and we used MoCHI to simultaneously fit a thermodynamic model to the molecular phenotypes of the variants in all seven experimental datasets (Extended Data Fig. 3b,c and Methods). Each single amino acid change in KRAS therefore has seven associated free energy changes: six binding energies and one folding energy (Fig. 4a and Extended Data Fig. 4a). As for RAF1 (Fig. 1k), the MoCHI binding energies for RALGDS correlate extremely well with independent in vitro measurements (Fig. 4b,c). The binding energies identify the known binding surfaces on KRAS, including the two known interfaces for SOS1 (ref. 31) (Fig. 2b and Extended Data Fig. 4b, median ROC-AUC = 0.80, range = 0.68–0.89 for weighted mean binding energies and median ROC-AUC = 0.64, range = 0.54-0.75 for weighted mean binding fitness measurements).

These seven free energy landscapes constitute more than 22,000 thermodynamic measurements, which is similar in scale to the number of measurements made for proteins in the entire scientific literature[32].

## Specificity in binding interfaces

We first considered how mutations in the binding interfaces alter binding to the six interaction partners. All six proteins bind KRAS through an overlapping set of contacts (Fig. 5a–c). This sharing of contacts is particularly pronounced for the three effector proteins, RAF1, PIK3CG and RALGDS (Fig. 5a). Comparing the mutational effects reveals that whereas some residues are critically important for binding to all three proteins, many substitutions alter the binding specificity (Fig. 5d). For example, many mutations in the negatively charged residues D33 and D38 and the hydrophobic residues I36 and Y40 strongly inhibit binding to all three effectors. However, a subset of hydrophobic substitutions at I36 inhibits binding to PIK3CG and RALGDS but not to RAF1 and substitution of L56 to negatively charged residues specifically increases

binding to RAF1 while retaining binding to PIK3CG but inhibiting binding to RALGDS (Fig. 5d). By contrast, many substitutions at E37 inhibit binding to RAF1 and RALGDS but increase binding to PIK3CG. Mutating Y64 inhibits binding to PIK3CG and RALGDS but allows binding to RAF1. At S39 a subset of hydrophobic mutations inhibit binding to PIK3CG and RAF1 but not to RALGDS. Comparing the binding free energies for all six binding partners reveals a high diversity of specificity changes that can be reached through single amino acid substitutions (Extended Data Fig. 5).

## Allosteric maps of six KRAS interactions

We next considered the specificity of mutational effects outside of the binding interfaces. We first focused on the positions that are most enriched for allosteric mutations for each interaction, defining the major allosteric sites for each interaction as those in which the average absolute binding free energy change is as large or greater than the average across mutations in all the binding interfaces (Fig. 6a). Novel major allosteric sites were identified for all six binding partners, with a median of 9 major allosteric sites in the nucleotide-binding pocket and a median of 5.5 additional major allosteric sites for each interaction (Fig. 6a).

We then compared the binding free energy changes between all six interaction partners for all mutations in these positions (Fig. 6b). Many substitutions at G10, G15, S17, D57, F78, P110 and V112 inhibit all six interactions (Fig. 6b and Extended Data Fig. 6a). Substitutions of F28 to non-aromatic amino acids inhibit all six interactions, as do many changes to charged amino acids at I55 and to hydrophobic amino acids at A18 and A83 (Fig. 6b). Substitutions to P at I55, A59, R68, K117 and F156 inhibit at least five interactions (Fig. 6b and Extended Data Fig. 6a). Considering all mutations outside of the binding interface, allosteric mutations are enriched at G, P, F and T residues for four out of six partners and depleted at charged residues for six out of six partners. Allosteric mutations are also enriched for substitutions to P for six out of six partners and to R for five out of six partners (Extended Data Fig. 6b). The enrichment for allosteric mutations at G residues and for substitutions to P is also observed in three small protein domains[13].

## Allosteric control of binding specificity

That multiple mutations at many of the allosteric sites inhibit binding to all interaction partners suggests that engagement of these sites is likely to generally inhibit KRAS function. However inspection of Fig. 6b also reveals sets of mutations in the major allosteric sites that have more specific allosteric effects. Particularly notable examples are many mutations in residues K16, I55, G60 and F156 that allosterically inhibit binding to most KRAS interaction partners but allosterically increase binding to the DARPin K27 (Fig. 6b). The DARPin K27 specifically recognizes inactive GDP-bound KRAS, and so mutations at these sites are likely to favour GDP-binding states. Consistent with this, K16 and G60 directly contact the γ-phosphate of GTP. Many substitutions of E76 also increase binding to DARPin K27 but with little effect on the other interactions. Additional examples include mutations at Y71 and M72 that specifically inhibit binding to DARPin K55 and mutations at D54 that inhibit four interactions but retain or enhance binding to PIK3CG and RALGDS (Fig. 6b). In addition, outside of these major allosteric sites there are many other mutations that allosterically alter both the binding affinity and specificity of KRAS (Extended Data Fig. 7).

## Discussion

Here we presented a global map of inhibitory allosteric sites for KRAS and a comprehensive comparative map of the effects of mutations on the free energies of binding of KRAS to multiple interaction partners.

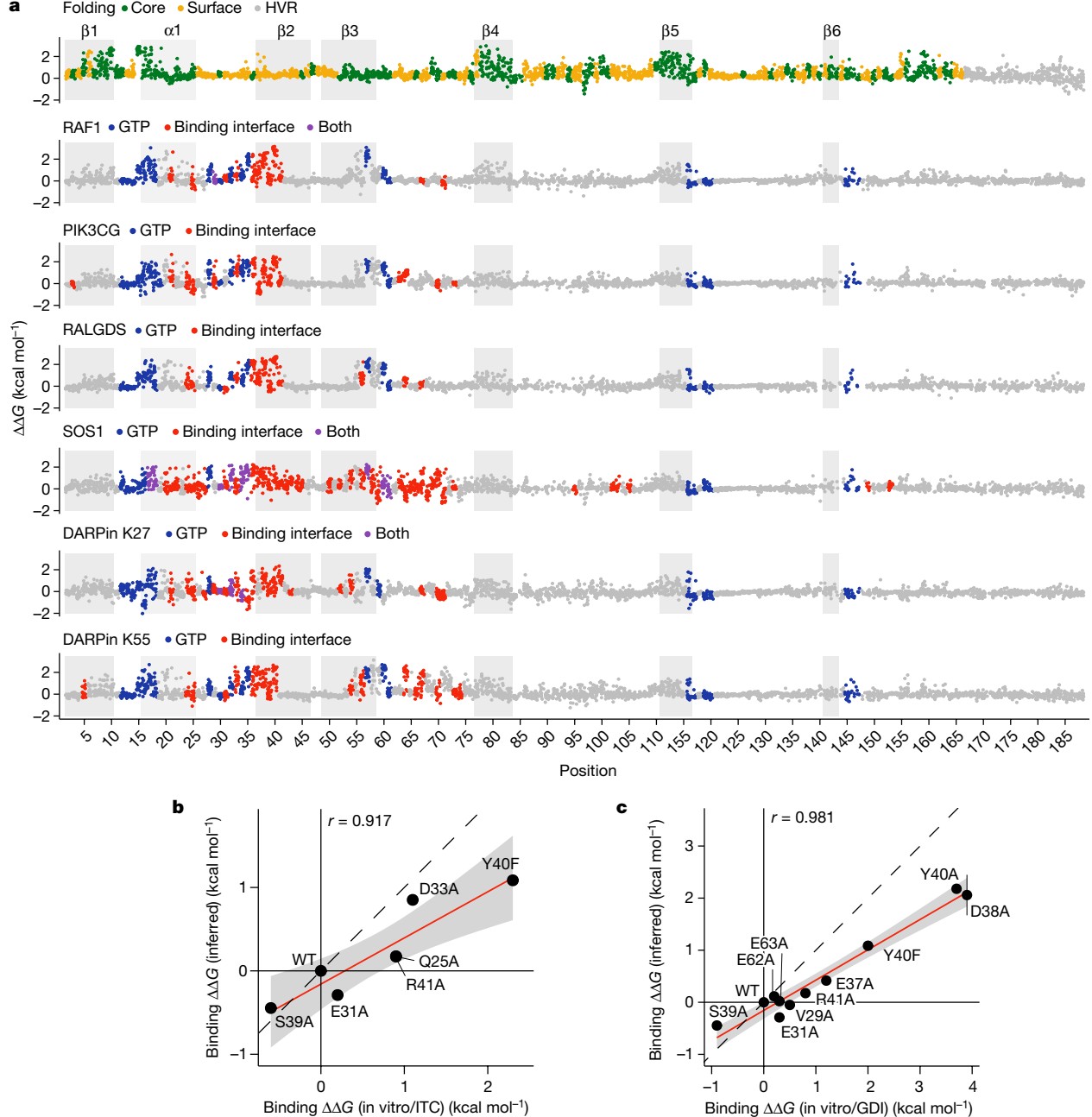

**Fig. 4 | Seven KRAS free energy landscapes. a**, Manhattan plots showing the folding and binding free energy changes of all single amino acid substitutions. Dark grey rectangles indicate β-strands, light grey rectangles indicate α-helix 1. HVR, hypervariable region. Binding interface is defined by indicated binding partner distance < 5 Å. PDB IDs: RAF1, 6VJJ; PIK3CG, 1HE8; RALGDS, 1LFD; SOS1, 1NVW; DARPin K27, 5O2S; DARPin K55, 5O2T. **b,c**, Comparisons of binding free energy changes to in vitro measurements by isothermal titration calorimetry (ITC) (**b**) and guanine nucleotide dissociation inhibition assay (GDI) (**c**). Linear regression fit and its 95% confidence interval are shown as a red solid line and a grey shaded area, respectively. Pearson's *r* is shown. Black dashed line indicates *y* = *x*. Error bars indicate 95% confidence intervals from a Monte Carlo simulation approach (*n* = 10 experiments).

The dataset constitutes more than 22,000 free energy measurements, a rich resource for protein biophysics and computational biology.

KRAS is one of the most frequently mutated genes in cancer and one of the most sought after and valuable therapeutic targets. Our results reveal a number of principles concerning allosteric communication in KRAS. First, KRAS has many inhibitory allosteric sites. Second, most allosteric mutations inhibit binding to all three KRAS effectors, revealing the potential to broadly inhibit KRAS activity. Third, allosteric mutations are enriched close to binding sites, suggesting local energetic propagation as the main allosteric mechanism. Fourth, allosteric communication is anisotropic, with communication being particularly effective across the central β-sheet of KRAS. Fifth, mutations can also allosterically control binding specificity, suggesting the potential for regulatory, evolutionary and therapeutic modulation of signalling bias. Sixth, all four surface pockets of KRAS are allosterically active, with particularly notable effects of mutations in the distal pocket 3. The comprehensive allosteric map therefore genetically validates all four pockets as suitable for therapeutic targeting and focuses attention on the largely ignored pocket 3.

The KRAS effector interface—similar to many protein surfaces—has to recognize structurally diverse proteins. Comprehensive mutagenesis of

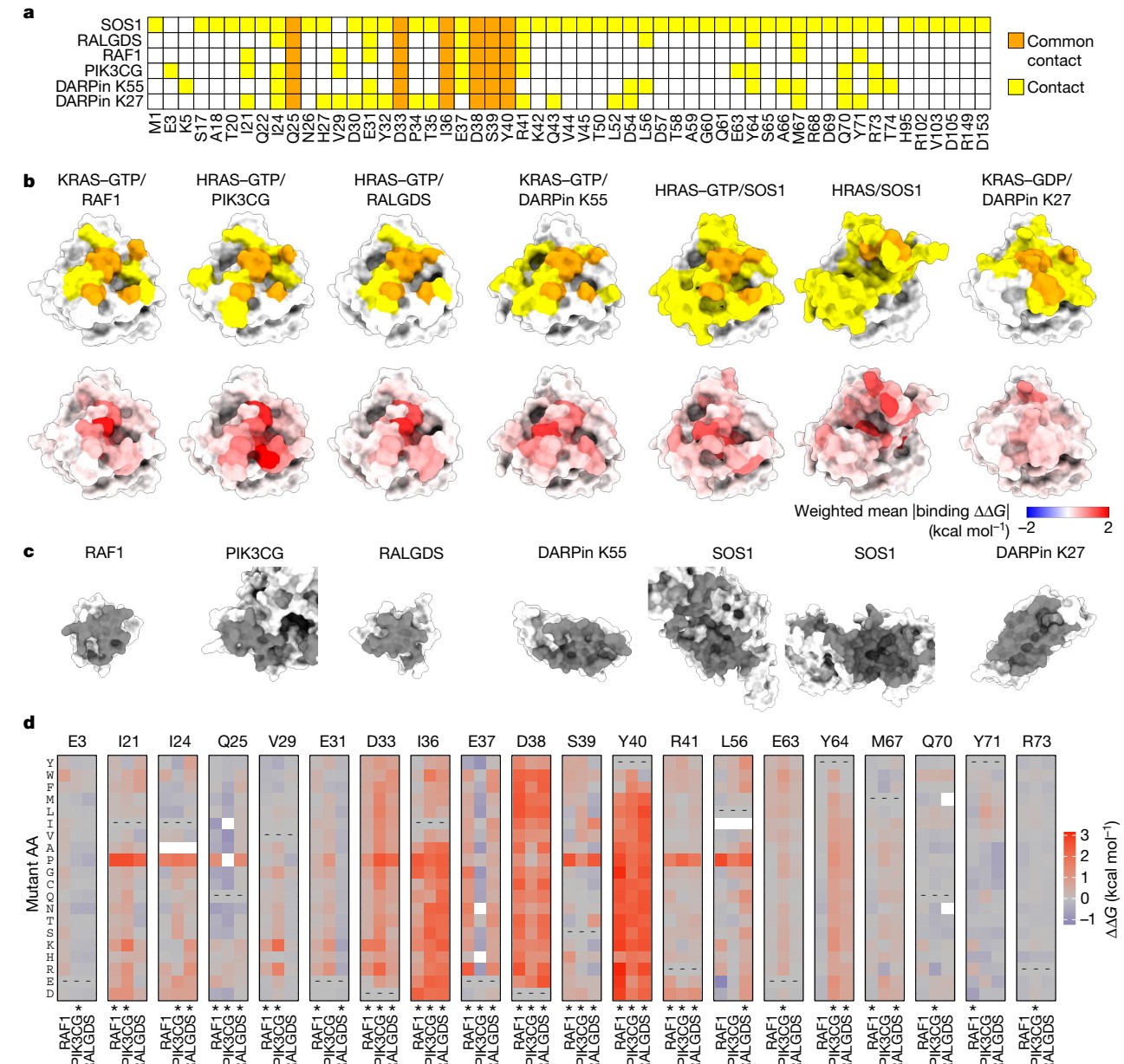

**Fig. 5 | Energetic landscapes of KRAS interaction surfaces. a**, Common and unique structural contacts between KRAS and the indicated binding partner. **b**, 3D structures of KRAS indicating binding partner contacts (top row, coloured as in **a**) and weighted mean absolute binding free energy change (bottom row). **c**, 3D structures of binding partners with binding interface indicated in grey.

PDB IDs: RAF1, 6VJJ; PIK3CG, 1HE8; RALGDS, 1LFD; DARPin K55, 5O2T; SOS1, 1NVW; DARPin K27, 5O2S. **d**, Heat maps of binding free energy changes in interface residues contacting at least one of the three effectors (RAF1, PIK3CG and RALGDS). Asterisks indicate binding interface residues for each partner.

this surface shows that its evolution is constrained by fitness trade-offs, with mutations that increase binding to one protein typically having antagonistic pleiotropic effects on binding to others. However, the binding specificity of KRAS is highly evolvable, with single amino acid substitutions causing a diversity of specificity changes. These altered binding profiles can be useful experimental tools, providing 'edgetic' perturbations[33] to test the functions of individual molecular interactions and their combinations[33,34].

In our experiments, we quantified mutational effects in wild-type KRAS. To test how well these effects are conserved in KRAS carrying oncogenic driver mutations, we reconstituted activation of RAF1 binding by driver mutations in yeast by co-expressing the catalytic domain of a human GAP, RASA1 (Extended Data Fig. 8a–i). Mutational effects in oncogenic KRAS were highly correlated to those in wild-type KRAS in

the absence and presence of human GAP co-expression (for example, Pearson's $r$ between wild-type KRAS and KRAS(G12C) in the presence of human GAP co-expression is 0.93, $n = 776$; Extended Data Fig. 8). A second potential caveat of our experiments was that we quantified binding of KRAS to isolated RBDs and, in general, mutations that have allosteric effects in isolated domains may have different effects or directly participate in binding in full-length proteins. However we found that changes in binding to full-length RAF1 were highly correlated to those to the RAF1 RBD (Pearson's $r = 0.94$, $n = 1,186$ genotypes), as were the inferred binding free energy changes (Pearson's $r = 0.89$, $n = 1,195$; Extended Data Fig. 9). Finally, we note that there are likely to be multiple molecular mechanisms that mediate the allosteric effects, including shifts in conformational equilibria, altered nucleotide binding or hydrolysis, and propagated structural and dynamic perturbations in the

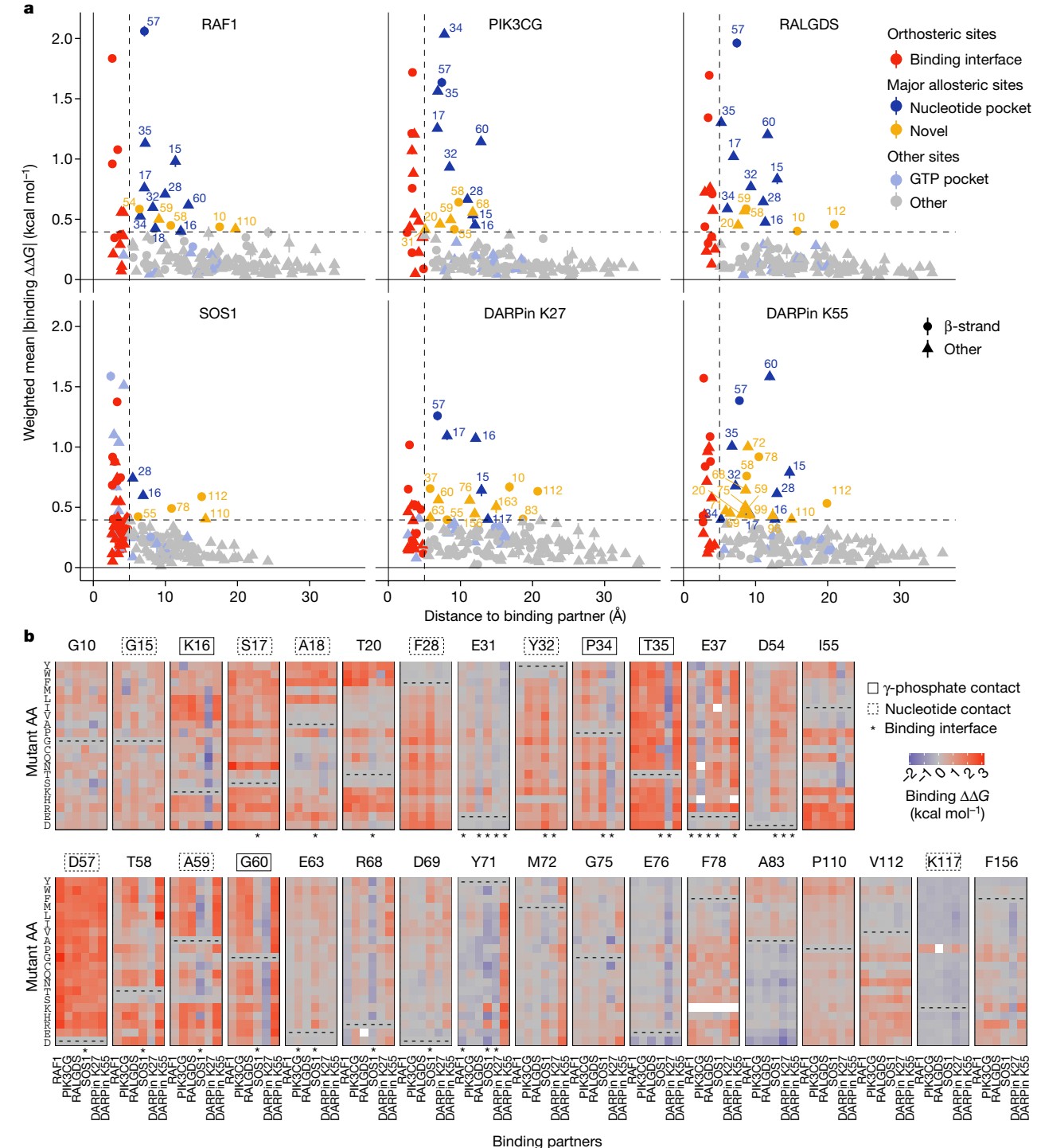

**Fig. 6 | Allosteric control of binding specificity. a**, Relationship between the weighted mean absolute change in free energy of binding and the distance to each corresponding binding partner (minimal side chain heavy atom distance). Major allosteric sites are defined as non-binding-interface residues with weighted mean absolute change in free energy of binding higher than the average of binding-interface-residue mutations across all binding partners (horizontal dashed line). Error bars indicate 95% confidence interval ($n \geq 10$). **b**, Heat maps of binding free energy changes in all major allosteric sites. Nucleotide pocket and γ-phosphate-contacting residues are indicated.

binding interfaces. Further experiments will be needed to disentangle the mechanistic causes of allostery.

The accelerated pace of human genetics research means we now know the proteins to therapeutically target in hundreds of human diseases[35]. However, effective therapies have been developed against a small minority of these genetically validated targets. In short, the protein targets for many diseases are known, but we do not know how

to target them. For most proteins, the location of the 'switches' to target with drugs to turn activity off or on remain unknown. If we could find these switches, we would be able to develop drugs to control their activity.

The data presented here and in other recent studies[13,36–39] have revealed that allosteric sites are much more prevalent than is widely appreciated. Moreover, the approach that we have applied here to

KRAS is quite general and can be used to identify allosteric sites in many different proteins. We believe that this general strategy can be used to systematically map regulatory sites that can be used to target many important proteins. Mapping of allosteric sites is likely to have an increasingly important role in drug development, laying the foundations for therapeutically targeting proteins that were previously considered to be undruggable.

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

# Methods

## Media and buffers

The following media and buffers were used and prepared as follows. Luria-Bertani (LB) medium: 10 g l$^{-1}$ Bacto-tryptone, 5 g l$^{-1}$ yeast extract, 10 g l$^{-1}$ NaCl; autoclaved 20 min at 120 °C. Yeast peptone dextrose adenine (YPDA): 20 g l$^{-1}$ glucose, 20 g l$^{-1}$ peptone, 10 g l$^{-1}$ yeast extract, 40 mg l$^{-1}$ adenine sulfate; autoclaved 20 min at 120 °C. Sorbitol medium (SORB): 1 M sorbitol, 100 mM lithium acetate, 10 mM Tris pH 8.0, 1 mM EDTA; filter sterilized (0.2 mm nylon membrane, Thermo Scientific). Plate mixture: 40% PEG3350, 100 mM lithium acetate, 10 mM Tris-HCl pH 8.0, 1 mM EDTA pH 8.0; filter sterilized. Recovery medium: yeast peptone dextrose (20 g l$^{-1}$ glucose, 20 g l$^{-1}$ peptone, 10 g l$^{-1}$ yeast extract), 0.5 M sorbitol. Filter sterilized. Synthetic complete medium without uracil (SC-URA): 6.7 g l$^{-1}$ yeast nitrogen base without amino acid, 20 g l$^{-1}$ glucose, 0.77 g l$^{-1}$ complete supplement mixture drop-out without uracil; filter sterilized. Synthetic complete medium without uracil, methionine and adenine (SC-URA/MET/ADE): 6.7 g l$^{-1}$ yeast nitrogen base without amino acid, 20 g l$^{-1}$ glucose, 0.74 g l$^{-1}$ complete supplement mixture drop-out without uracil, adenine and methionine; filter sterilized. Competition medium: SC-URA/MET/ADE + 200 µg ml$^{-1}$ methotrexate (BioShop Canada), 2% DMSO. DNA extraction buffer: 2% Triton X-100, 1% SDS, 100 mM NaCl, 10 mM Tris-HCl pH 8, 1 mM EDTA pH 8.

## Plasmid construction

Two generic plasmids were constructed to be able to assay any protein of interest by BindingPCA or AbundancePCA: the BindingPCA plasmid (pGJJ161) and the AbundancePCA plasmid (pGJJ162).

The BindingPCA plasmid (pGJJ161) and AbundancePCA plasmid (pGJJ162) were derived from the previous BindingPCA plasmid (pGJJ001) and the previous AbundancePCA plasmid (pGJJ045)[13]. The C-terminus (GGGGS)4 linker of DHFR3 were changed to the N terminus, which allowed us to fuse the protein of interest to the N terminus of the DHFR3 fragment in both abundance and BindingPCA assays.

One KRAS AbundancePCA plasmid, 7 BindingPCA plasmids, one BindingPCA co-expression RASGAP (the catalytic domain of human RASA1) plasmid and one KRAS mutagenesis plasmid are used in this paper. To construct the KRAS AbundancePCA plasmid (pGJJ271), the sequence of full-length KRAS (188 amino acids) was amplified from a plasmid, a gift from the L. Serrano laboratory, using primer pair oGJJ231/oGJJ232 (Supplementary Table 1). This primer pair also introduced the HindIII and NheI restriction sites. The PCR product was digested by HindIII and NheI, then was cloned into the digested pGJJ162 plasmid using T4 Ligase (NEB). To construct 7 KRAS BindingPCA plasmids, a common KRAS BindingPCA plasmid (pGJJ317) was constructed by ligating full-length KRAS sequence digested by HindIII and NheI to digested BindingPCA plasmid. 7 BindingPCA plasmids are constructed by ligating each binding partners PCR product which was digested by BamHI and SpeI to digested pGJJ317 using T4 Ligase (NEB). To construct RAF1 BindingPCA plasmid (pGJJ336), the sequence of RAF1-RBD (52–131) was amplified from the cDNA of 293 T cell line using primer pair oGJJ74/oGJJ307 which also introduced the BamHI and SpeI restriction sites. To construct PI3KCG BindingPCA plasmid (pGJJ565), the sequence of PIK3CG RBD (203–312) was amplified from R777-E169 Hs.PIK3CG (Addgene) using primer pair oWCC169/oWCC170. To construct RALGDS BindingPCA plasmid (pGJJ400), the sequence of RALGDS RBD (778–864) was amplified from R777-E169 Hs.PIK3CG (Addgene) using primer pair oWCC28/oWCC29. To construct SOS1 BindingPCA plasmid (pGJJ541), the sequence of SOS1 (564–1049) was amplified from plasmid R777-E317 Hs.SOS1 (Addgene) using primer pair oWCC149/oWCC150. To construct DARPin K27 BindingPCA plasmid (pGJJ553), the sequence of DARPin K27 was amplified from plasmid pCASP-SptP120-K27-HilA (Addgene) using primer pair oWCC157/oWCC158. To construct DARPin K55 BindingPCA plasmid (pGJJ554), the sequence of DARPin K55 was amplified from plasmid pCASP-SptP120-K55-HilA (Addgene) using primer pair oWCC159/oWCC160. To construct full-length RAF1 BindingPCA plasmid (pGJJ623), the sequence of full-length RAF1 (amino acids 1 to 648) was amplified from a gene block synthesized by IDT (Integrated DNA Technologies) using primer pair oWCC252/oWCC253. To construct the BindingPCA co-expression RASGAP plasmid, the *cyc1* promoter-driven RASGAP cassette was amplified in four fragments, *cyc1* promoter from AbundancePCA plasmid (pGJJ271) using primer pair oWCC182/oWCC183, two fragments of RASGAP (amino acids 714 to 1047) were amplified from ORFeome plasmid (81020C02, Protein Technologies Unit, CRG) using primer pairs oWCC184/oWCC97, and oWCC96/oWCC129, *cyc1* terminator was amplified from AbundancePCA plasmid (pGJJ271) using primer pair oWCC128/oWCC140, which were then assembled by Gibson reaction (Protein Technologies Unit, CRG) at 50 °C for 1 h with RAF1-RBD BindingPCA plasmid (pGJJ336) which was digested by NgoMIV. To construct the KRAS mutagenesis plasmid (pGJJ380), pGJJ191 plasmid was constructed firstly which contained a streptomycin resistance gene cassette. The pGJJ191 plasmid was amplified in two fragments: one ori cassette which also contained AvrII and HindIII restriction sites using primer pair oGJJ308/oGJJ309, the other streptomycin resistance gene cassette using primer pair oGJJ310/oGJJ311, which were then assembled by Gibson reaction at 50 °C for 1 h. The KRAS sequence was digested by AvrII and HindIII from AbundancePCA plasmid and ligated into digested pGJJ191. Then a BbvCI restriction site was introduced using primer pair oWCC51/oWCC52.

## Mutagenesis library construction

The plasmid-based one-pot saturation (nicking) mutagenesis protocol was used in this study[14]. KRAS are divided to three blocks in order to be fully sequenced by Illumina paired-end 150 NextSeq pipeline.

An initial single round of nicking mutagenesis using equimolar mixes of degenerate KRAS primers (Supplementary Table 2) was obtained for two reasons: (1) To obtain random single mutants to use as template for another round of nicking mutagenesis (by randomly selecting single colonies and verified by Sanger sequencing); and (2) to quantify the degenerate primer positional bias and compensate for it in the shallow double mutant libraries.

To construct three final KRAS libraries, an equimolar pool of single mutants of each block and wild type were used as the plasmid template for a round of nicking mutagenesis. The mutants were chosen based on their varying binding affinities to RAF1 (refs. 18,19), ensuring a range of affinities within the mutant pools (block 1: T2K, V14S, L6H, E37G, Y40A, D38C, L19P, Q61L, E63V; block 2: I84L, F82S, L113F, Y71F, K101R, A66P, M72G, F78W, E63V, V112N; block 3: K176C, R149V, L133A, Y137K, L159A and A146F). Additionally, the mutants of interest (G12C, G12D, G12V, S17N and T35S) were also included in block 1. To compensate for the extreme positional biases, each mutagenic primer was mixed in the pool inversely to the mean read counts per position from these first-round nicking libraries.

The libraries midi preps were digested with HindIII and NheI restriction enzymes and the insert containing the mutated protein was gel purified (MinElute Gel Extraction Kit, QIAGEN) to be later cloned into the AbundancePCA plasmid and BindingPCA plasmids by temperature-cycle ligation. The AbundancePCA plasmid and BindingPCA plasmids were all digested by HindIII and NheI enzymes and purified using the QIAquick Gel Extraction Kit (QIAGEN). The assembly of AbundancePCA libraries and BindingPCA libraries were done overnight by temperature-cycle ligation using T4 ligase (New England Biolabs) according to the manufacturer's protocol, 67 fmol of backbone and 200 fmol of insert in a 33.3 µl reaction. The ligation was desalted by dialysis using membrane filters for 1 h and later concentrated 3.3× using a SpeedVac concentrator (Thermo Scientific).

All concentrated assembled libraries were transformed into NEB 10β High-efficiency Electrocompetent *Escherichia coli* cells according to the manufacturer's protocol (volumes used in each library specified in

Supplementary Table 3). Cells were allowed to recover in SOC medium (NEB 10β Stable Outgrowth Medium) for 30 min and later transferred to 200 ml LB medium with ampicillin 4× overnight. The total number of estimated transformants for each library can be found in Supplementary Table 3. One-hundred millilitres of each saturated *E. coli* culture were collected next morning to extract the plasmid library using the QIAfilter Plasmid Midi Kit (QIAGEN).

## Methotrexate selection assays

The methotrexate selection assay protocol was described in a previous study[13]. The high-efficiency yeast transformation protocol was scaled in volume depending on the targeted number of transformants of each library. The transformation protocol described below (adjusted to a pre-culture of 175 ml of YPDA) was scaled up or down in volume as reported in Supplementary Table 3.

For each of the selection assays (3 blocks × 6 BindingPCA + 3 blocks × 1 AbundancePCA), 3 independent pre-cultures of BY4742 were grown in 20 ml standard YPDA at 30 °C overnight. The next morning, the cultures were diluted into 175 ml of pre-warmed YPDA at an optical density at 600 nm ($OD_{600}$) of 0.3. The cultures were incubated at 30 °C for 4 h. After growth, the cells were collected and centrifuged for 5 min at 3,000 $g$, washed with sterile water and later with SORB medium (100 mM lithium acetate, 10 mM Tris pH 8.0, 1 mM EDTA, 1 M sorbitol). The cells were resuspended in 8.6 ml of SORB and incubated at room temperature for 30 min. After incubation, 175 μl of 10 mg ml⁻¹ boiled salmon sperm DNA (Agilent Genomics) was added to each tube of cells, as well as 3.5 μg of plasmid library. After gentle mixing, 35 ml of plate mixture was added to each tube to be incubated at room temperature for a further 30 min. DMSO (3.5 ml) was added to each tube and the cells were then heat shocked at 42 °C for 20 min (inverting tubes from time to time to ensure homogeneous heat transfer). After heat shock, cells were centrifuged and resuspended in ~50 ml of recovery medium and allowed to recover for 1 h at 30 °C. Next, cells were again centrifuged, washed with SC-URA medium and resuspended in SC-URA (volume used in each library found in Supplementary Table 3). After homogenization by stirring, 10 μl were plated on SC-URA petri dishes and incubated for ~48 h at 30 °C to measure the transformation efficiency. The independent liquid cultures were grown at 30 °C for ~48 h until saturation. The number of yeast transformants obtained in each library assay can be found in Supplementary Table 3.

For each of the BindingPCA or AbundancePCA assays, each of the growth competitions was performed right after yeast transformation. After the first cycle of post-transformation plasmid selection, a second plasmid selection cycle (input) was performed by inoculating SC-URA/MET/ADE at a starting $OD_{600}$ = 0.1 with the saturated culture (volume of each experiment specified in Supplementary Table 3). Cells were grown for 4 generations at 30 °C under constant agitation at 200 rpm (selection time of each experiment specified in Supplementary Table 3). This allowed the pool of mutants to be amplified and enter the exponential growth phase. The competition cycle (output) was then started by inoculating cells from the input cycle into the competition medium (SC-URA/MET/ADE + 200 μg ml⁻¹ methotrexate) so that the starting $OD_{600}$ was 0.05. For that, the adequate volume of cells was collected, centrifuged at 3,000 rpm for 5 min and resuspended in the pre-warmed output medium. Meanwhile, each input replicate culture was split in two and collected by centrifugation for 5 min at 5,000 $g$ at 4 °C. Yeast cells were washed with water, pelleted and stored at −20 °C for later DNA extraction. After ~4 generations of competition cycles, each output replicate culture was split in two and collected by centrifugation for 5 min at 5,000 $g$ at 4 °C, washed twice with water and pelleted to be stored at −20 °C.

## DNA extractions and plasmid quantification

The DNA extraction protocol used was described previously[13]. A 50 ml collected culture of $OD_{600}$ ≈ 1.6 is described below. Cell pellets (one for each experiment input or output replicate) were resuspended in 1 ml of DNA extraction buffer, frozen by dry ice-ethanol bath and incubated at 62 °C water bath twice. Subsequently, 1 ml of phenol:chloroform:isoamyl alcohol 25:24:1 (equilibrated in 10 mM Tris-HCl, 1 mM EDTA, pH 8) was added, together with 1 g of acid-washed glass beads (Sigma Aldrich) and the samples were vortexed for 10 min. Samples were centrifuged at room temperature for 30 min at 4,000 rpm and the aqueous phase was transferred into new tubes. The same step was repeated twice. Three molar sodium acetate (0.1 ml) and 2.2 ml of pre-chilled absolute ethanol were added to the aqueous phase. The samples were gently mixed and incubated at −20 °C for 30 min. After that, they were centrifuged for 30 min at full speed at 4 °C to precipitate the DNA. The ethanol was removed and the DNA pellet was allowed to dry overnight at room temperature. DNA pellets were resuspended in 0.6 ml TE 1× and treated with 5 μl of RNaseA (10 mg ml⁻¹, Thermo Scientific) for 30 min at 37 °C. To desalt and concentrate the DNA solutions, QIAEX II Gel Extraction Kit was used (50 μl of QIAEX II beads). The samples were washed twice with PE buffer and eluted twice by 125 μl of 10 mM Tris-HCl buffer, pH 8.5 and then the two elutions were combined. Finally, plasmid concentrations in the total DNA extract (that also contained yeast genomic DNA) were quantified by quantitative PCR using the primer pair oGJJ152/oGJJ153, that binds to the ori region of the plasmids.

## Sequencing library preparation

The sequencing library preparation protocol was described previously[13]. The sequencing libraries were constructed in two consecutive PCR reactions. The first PCR (PCR1) was designed to amplify the mutated protein of interest and to increase the nucleotide complexity of the first sequenced bases by introducing frame-shift bases between the adapters and the sequencing region of interest. The second PCR (PCR2) was necessary to add the remainder of the Illumina adapter and demultiplexing indexes.

To avoid PCR biases, PCR1 of each independent sample (input/output replicates of any of the yeast assays) was run with an excess of plasmid template 20–50 times higher than the number of expected sequencing reads per sample. Each reaction started with a maximum of $1.25 × 10^7$ template plasmid molecules per microlitre of PCR1, avoiding introducing more yeast genomic DNA that interfered with the efficiency of the PCR reaction. For this reason, PCR1s were scaled up in volume as specified in Supplementary Table 3. The PCR1 reactions were run using Q5 Hot Start High-Fidelity DNA Polymerase (New England Biolabs) according to the manufacturer's protocol, with 25 pmol of pooled frame-shift primers as specified in Supplementary Table 1 for different blocks (forward and reverse primers were independently pooled according to the nucleotide diversity of each oligonucleotide, Supplementary Table 1). The PCR reactions were set to 60 °C annealing temperature, 10 s of extension time and run for 15 cycles. Excess primers were removed by adding 0.04 μl of ExoSAP-IT (Affymetrix) per microlitre of PCR1 reaction and incubated for 20 min at 37 °C followed by an inactivation for 15 min at 80 °C. The PCRs of each sample were then pooled and purified using the MinElute PCR Purification Kit (QIAGEN) according to the manufacturer's protocol. DNA was eluted in EB to a volume six times lower than the total volume of PCR1.

PCR2 reactions were run for each sample independently using Hot Start High-Fidelity DNA Polymerase. The total reaction of PCR2 was reduced to half of PCR1, using 0.05 μl of the previous purified PCR1 per microlitre of PCR2. In this second PCR the remaining parts of the Illumina adapters were added to the library amplicon. The forward primer (5′ P5 Illumina adapter) was the same for all samples, while the reverse primer (3′ P7 Illumina adapter) differed by the barcode index (oligonucleotide sequences in Supplementary Table 1), to be subsequently pooled together and demultiplex them after deep sequencing (indexes used in each replicate of each sequencing run found in Supplementary Table 3). Eight cycles of PCR2s were run at 62 °C of

annealing temperature and 10 s of extension time. All reactions from the same sample were pooled together and an aliquot was run on a 2% agarose gel to be quantified. All samples were purified using the QIAEX II Gel Extraction Kit. The purified amplicon library pools were subjected to Illumina 150 bp paired-end NextSeq sequencing at the CRG Genomics Core Facility.

## Sequencing data processing

FastQ files from paired-end sequencing of all BindingPCA and AbundancePCA experiments were processed with DiMSum v1.2.9 (ref. 41) (https://github.com/lehner-lab/DiMSum) using default settings with minor adjustments. Supplementary Table 4 contains DiMSum fitness estimates and associated errors for all experiments. Experimental design files and command-line options required for running DiMSum on these datasets are available on GitHub (https://github.com/lehner-lab/krasddpcams). In all cases, adaptive minimum Input read count thresholds based on the corresponding number of nucleotide substitutions ('fitnessMinInputCountAny' option) were selected in order to minimize the fraction of reads per variant related to sequencing error-induced 'variant flow' from lower order mutants.

Variant counts associated with all samples (output from DiMSum stage 4) were further filtered using a custom script to retain only those variants with single amino acid substitutions including a G/T in the third codon position (encoded by NNK) or amino acid substitutions representing high confidence backgrounds. The latter were defined as single amino acid substitutions observed at least 200 times (in different double amino acid variants) in at least five (out of a total of seven) BindingPCA/AbundancePCA experiments. For double amino acid variants, we required one of the constituent single amino acid variants to be a high confidence background mutation. All read counts associated with remaining single or double amino acid variants (probably the result of PCR and sequencing errors) were discarded. Finally, fitness estimates and associated errors were then obtained from the resulting filtered variant counts with DiMSum (countPath option).

## Thermodynamic model fitting with MoCHI

We used MoCHI v0.9 (https://github.com/lehner-lab/MoCHI)[22] to fit a global mechanistic model to all 21 ddPCA datasets (7 phenotypes × 3 blocks) simultaneously. The software is based on our previously described genotype–phenotype modelling approach[13] with additional functionality and improvements for ease of use and flexibility.

In brief, we model individual KRAS PPIs as an equilibrium between three states: unfolded and unbound (uu), folded and unbound (fu), and folded and bound (fb). We assume that the probability of the unfolded and bound state (ub) is negligible and free energies of folding and binding are additive—that is, the total binding and folding free energy changes of an arbitrary variant relative to the wild-type sequence is simply the sum over residue-specific energies corresponding to all constituent single amino acid substitutions. Furthermore, we assume binding energies are specific for each binding partner whereas folding energies are shared or intrinsic to KRAS—that is, unaffected by the identity, presence or expression of a given binding partner. We also assume that mutation effects on abundance level predominantly arise from folding free energy changes. However, protein abundance can be influenced by factors beyond folding, such as degradation or cellular processes, which may skew the free energy estimates.

We configured MoCHI parameters to specify a neural network architecture consisting of seven additive trait layers (free energies)—that is, one for each biophysical trait to be inferred (6 binding and 1 folding), as well as one linear transformation layer per experiment (3 AbundancePCA and 18 BindingPCA fitness). The specified nonlinear transformations 'TwoStateFractionFolded' and 'ThreeStateFractionBound' derived from the Boltzmann distribution function relate energies to proportions of folded and bound molecules respectively. The target (output) data to fit the neural network comprises fitness scores for wild-type, single and double amino acid substitution variants from all 21 ddPCA datasets.

A random 30% of double amino acid substitution variants was held out during model training, with 20% representing the validation data and 10% representing the test data. Validation data were used to evaluate training progress and optimize hyperparameters (batch size). Optimal hyperparameters were defined as those resulting in the smallest validation loss after 100 training epochs. Test data were used to assess final model performance.

MoCHI optimizes the parameters $\theta$ of the neural network using stochastic gradient descent on a loss function $L[\theta]$ based on a weighted and regularized form of mean absolute error:

$$L[\theta] = 1/N \sum_{n=0}^{N-1} |y_n - \hat{y}_n| \ \sigma_n^{-1} + \lambda_2 \ ||\theta||^2$$

where $y_n$ and $\sigma_n$ are the observed fitness score and associated standard error respectively for variant $n$, $\hat{y}_n$ is the predicted fitness score, $N$ is the batch size and $\lambda_2$ is the $L_2$ regularization penalty. In order to penalize very large free energy changes (typically associated with extreme fitness scores) we set $\lambda_2$ to $10^{-6}$ representing light regularization. The mean absolute error is weighted by the inverse of the fitness error ($\sigma_n^{-1}$) in order to downweight the contribution of less confidently estimated fitness scores to the loss. Furthermore, in order to capture the uncertainty in ddPCA fitness estimates, the training data were replaced with a random sample from the fitness error distribution of each variant. The validation and test data were left unaltered.

Models were trained with default settings—that is, for a maximum of 1,000 epochs using the Adam optimization algorithm with an initial learning rate of 0.05. MoCHI reduces the learning rate exponentially ($\gamma = 0.98$) if the validation loss has not improved in the most recent ten epochs compared to the preceding ten epochs. In addition, MoCHI stops model training early if the wild-type free energy terms over the most recent ten epochs have stabilized (standard deviation ≤$10^{-3}$).

Free energies are calculated directly from model parameters as follows: $\Delta G_b = \theta b RT$ and $\Delta G_f = \theta f RT$, where $T = 303$ K and $R = 0.001987$ kcal K$^{-1}$ mol$^{-1}$. We estimated the confidence intervals of model-inferred free energies using a Monte Carlo simulation approach. The variability of inferred free energy changes was calculated between ten separate models fit using data from (1) independent random training–validation–test splits; and (2) independent random samples of fitness estimates from their underlying error distributions. Confident inferred free energy changes are defined as those with Monte Carlo simulation derived 95% confidence intervals of less than 1 kcal mol$^{-1}$. Supplementary Table 5 contains inferred binding and folding free energy changes of mutations for all binding partners.

## Recombinant protein sample preparation

KRAS residues 1–169 fused to an N-terminal His$_6$ tag and a TEV protease cleavage site was cloned into a pCoofy31 vector, and variants were generated by using the Q5 site-directed mutagenesis kit (New England Biolabs). Vectors were transformed into *E. coli* BL21 competent cells (NEB), and single colonies were picked to grow overnight pre-cultures to saturation in Luria-Bertani broth (LB) containing 33 μg ml$^{-1}$ kanamycin. Ten millilitres of the pre-cultures were used to inoculate antibiotic-supplemented 1 l LB cultures, which were grown at 24 °C to OD$_{600}$ ≈ 0.4, then at 18 °C to OD$_{600}$ ≈ 0.6. Protein expression was induced with 0.5 mM isopropyl β-D-1-thiogalactopyranoside (IPTG), and induced cultures were grown at 18 °C overnight. Cells were collected by centrifugation (15 min, 3,000g, 4 °C), resuspended in KRAS lysis buffer (20 mM Tris, 500 mM NaCl, 25 mM imidazole, 5 mM MgCl2, 2 mM β-mercaptoethanol, pH 8) supplemented with one tablet of Pierce protease inhibitor tablets, 0.5 mg ml$^{-1}$ PMSF (both from ThermoFisher), 0.1 mg ml$^{-1}$ bovine pancreas DNAse I and 1.5 mg ml$^{-1}$ chicken egg white lysozyme (both from Sigma Aldrich), and lysed in an Emulsiflex-C5

homogenizer (Avestin) at a maximum pressure of 1,500 psi. Cell debris was removed by ultracentrifugation (20 min, 40,000g, 4 °C) and the cleared lysate was loaded on a His-Trap Fast Flow column mounted on an Äkta Pure chromatography system (both from Cytiva). Column-bound recombinant KRAS variants were washed with KRAS lysis buffer containing 1 M KCl and eluted over a 15-column-volume gradient with lysis buffer containing 0.5 M imidazole. Collected 0.25 ml fractions were analysed by SDS–PAGE, pooled based on purity and concentrated using Amicon 10 kDa MWCO centrifugal filters (Merck Millipore).

Nucleotide exchange to load the non-hydrolysable GTP analogue guanosine 5′-[β,γ-imido]triphosphate (GppNHp, Sigma Aldrich) was achieved by adapting a previously detailed method[42]. In brief, concentrated KRAS variants were diluted to a concentration of 1.8 mg ml$^{-1}$ and a final volume of 2.5 ml in GppNHp loading buffer (50 mM Tris, 200 mM $(NH_4)_2SO_4$, 2 mM β-mercaptoethanol, pH 8) containing 3 mg of GppNHp. After 1 h incubation at 4 °C in a rotating wheel, samples were passed through a PD-10 column and eluted with 3.5 ml of GppNHp loading buffer. 30 units (6 µl) of QuickCIP (NEB) were added along with 2 mg of GppNHp, and samples were incubated for an additional 1 h on a rotating wheel at 4 °C. Subsequently, $MgCl_2$ was added to a concentration of 30 mM.

Both GDP and GppNHp-loaded samples were concentrated down to 0.5 ml and injected to a Superdex 75 10/300 GL column (Cytiva) equilibrated with SPR buffer (20 mM HEPES, 150 mM NaCl, 1 mM TCEP, pH 7.4) and mounted on an Äkta Pure system for size-exclusion chromatography. 0.5 ml fraction purity was assessed by SDS–PAGE, and fractions with ≥95% purity were flash-frozen in liquid nitrogen and stored at −80 °C until required for SPR measurement.

RAF1 residues 56–131, as well as DARPin K55 residues 1–156, fused to an N-terminal Twin-Strep tag and a 3 C HRV protease cleavage site were also cloned into pCoofy31. Inoculated cultures were grown at 37 °C to $OD_{600} \approx 0.6$, induced with 1 mM IPTG, and collected after 3 h growth at 37 °C. Cleared lysates in ligand lysis buffer (100 mM Tris, 150 mM NaCl, 1 mM EDTA, pH 8) were loaded on a StrepTrap XT prepacked chromatography column mounted on an Äkta Pure system (Cytiva). Bound protein was step-eluted with ligand lysis buffer containing 50 mM biotin, fractions were pooled based on SDS–PAGE-assessed purity and concentrated using 10 kDa MWCO centrifugal filters. Size-exclusion chromatography in SPR buffer and storage were performed in an analogous manner as described above for KRAS.

## Surface plasmon resonance

Samples were thawed on ice, centrifuged at 13,000g for 10 min, transferred to a new tube, and quantified using a NanoDrop One (Thermo-Fisher). Binding kinetics and affinity of KRAS variants for RAS or K55 were evaluated by surface plasmon resonance on a BIAcore T200 instrument (Cytiva) with SPR running buffer (10 mM HEPES, 150 mM NaCl, 0.05% Tween 20, pH 7.2). The assay format involved a Series S CM5 chip functionalized with Streptactin (50 µg ml$^{-1}$). In brief, amine coupling was used to create a Streptactin surface (Strep-Tactin XT) following instructions provided with the Twin-Strep-tag capture kit (IBA Lifesciences). Twin-Strep-tagged RAS or K55 protein constructs

were captured on flow cell 4, leaving flow cell 3 as a subtractive reference. Capture levels of RAS or K55 were targeted between 50 and 100 resonance units, after which increasingly concentrated samples of KRAS variants were flowed over immobilized RAS or K55 (50 µl min$^{-1}$ for 1 min) and allowed to dissociate up to 3 min. A concentration series of each KRAS variant ranging from 0.74 nM to 60 nM was used to analyse binding to RAS or K55. The capture surface was regenerated with a 60 s injection of 3 M guanidine hydrochloride (50 µl min$^{-1}$ for 1 min). All sensograms were analysed using a 1:1 Langmuir binding model.

## Reporting summary

Further information on research design is available in the Nature Portfolio Reporting Summary linked to this article.

## Data availability

All DNA sequencing data have been deposited in the Sequence Read Archive (SRA) under BioProject PRJNA907205. All fitness measurements and free energies are provided in Supplementary Tables 4 and 5 and released on MAVEdb (MAVEdb accession: urn:mavedb:00000115).

## Code availability

Source code for fitting thermodynamic models (MoCHI) is available at https://github.com/lehner-lab/MoCHI. Source code for all downstream analyses and to reproduce all figures described here is available at https://github.com/lehner-lab/krasddpcams.

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

**Acknowledgements** This work was funded by European Research Council (ERC) Advanced grant (883742), the Spanish Ministry of Science and Innovation (LCF/PR/HR21/52410004, EMBL Partnership, Severo Ochoa Centre of Excellence), the Bettencourt Schueller Foundation, the AXA Research Fund, Agencia de Gestio d'Ajuts Universitaris i de Recerca (AGAUR, 2017 SGR 1322), and the CERCA Program/Generalitat de Catalunya. C.W. was funded by an EMBO long-term fellowship (ALTF 881-2020). A.J.F. was funded by a Ramón y Cajal fellowship (RYC2021-033375-I) financed by the Spanish Ministry of Science and Innovation (MCIN/AEI/ 10.13039/501100011033) and the European Union (NextGenerationEU/PRTR). We thank all members of the Lehner laboratory for helpful discussions and suggestions. We thank the CRG Genomics Unit for sequencing and the Protein Technologies Unit for assistance with the SPR measurements.

**Author contributions** B.L. and C.W. conceived the project and designed the experiments. C.W. and A.E. performed the experiments. C.W. and A.J.F. performed the data analysis. B.L., C.W., A.J.F. and A.E. wrote the manuscript.

**Competing interests** A.J.F. and B.L. are founders, employees and shareholders of ALLOX.

**Additional information**
**Correspondence and requests for materials** should be addressed to Ben Lehner.

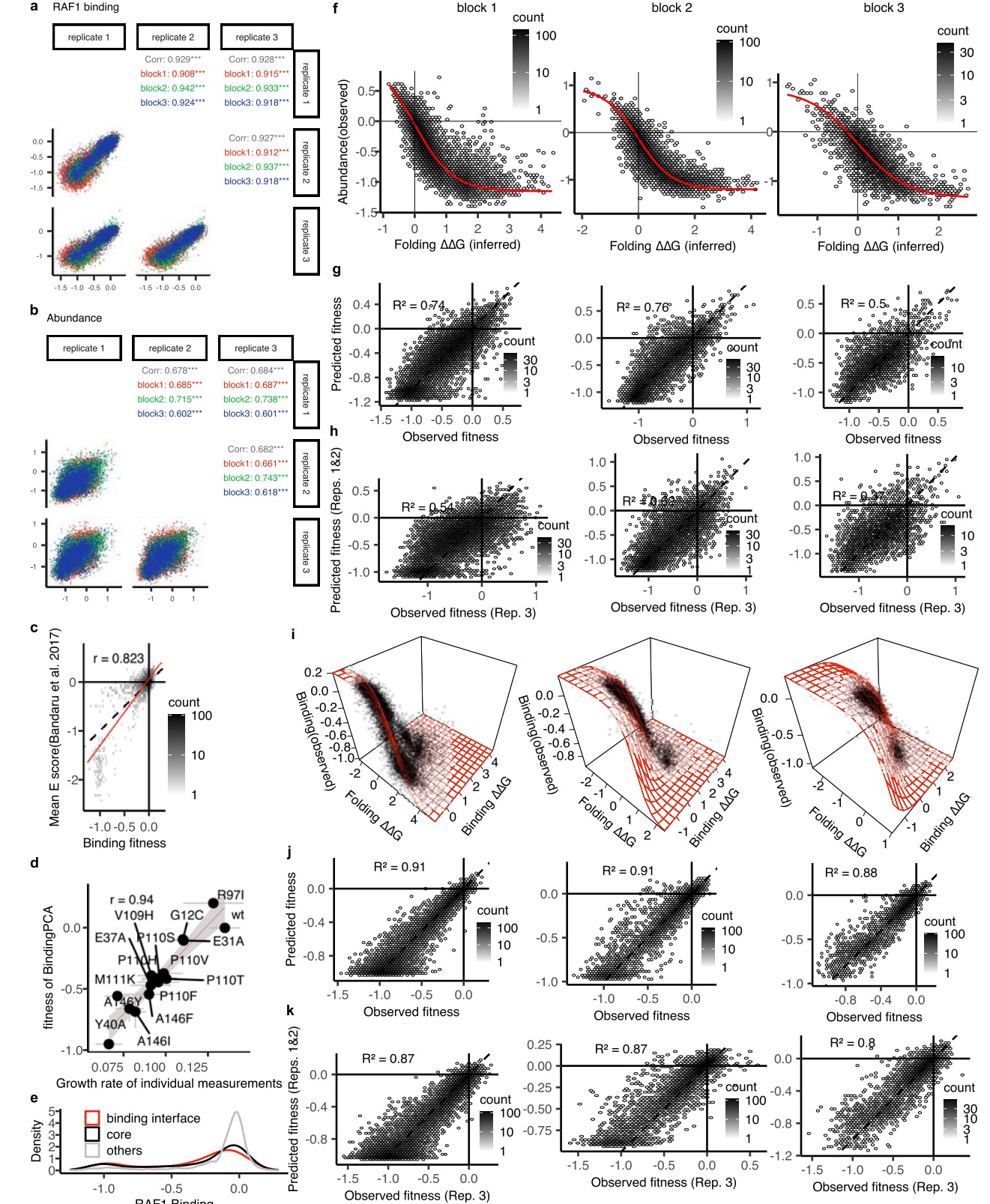

**Extended Data Fig. 1** | See next page for caption.

**Extended Data Fig. 1 | Experimental reproducibility and thermodynamic model fitting. a,b,** Scatter plots showing the reproducibility of each block's RAF1 BindingPCA (**a**) and AbundancePCA (**b**) fitness estimates from ddPCA. Pearson's *r* indicated on the top right corner. **c,** Comparison of RAF1 BindingPCA fitness to previously reported KRAS-RAF1 binding E score[19]. Pearson's *r* = 0.82. **d,** Comparison of individually measured growth rates to their corresponding fitness from deep sequencing for KRAS. The red line corresponds to a linear regression model. Pearson's *r* is shown. **e,** Single mutation fitness density distributions. **f,** 2D density plots showing non-linear relationships (global epistasis) between observed AbundancePCA fitness and changes in free energy of folding. **g,** 2D density plots comparing model predictions and observations of AbundancePCA fitness for held out test data (comprising 10% of double aa substitution variants held out during model training). **h,** Same as (**g**) except the model was trained using data from replicates 1 and 2 and evaluated using data from replicate 3. **i,** Non-linear relationships (global epistasis) between observed BindingPCA fitness and both free energies of binding and folding. **j,** 2D density plots comparing model predictions and observations of BindingPCA fitness for held out test data. **k,** Same as (**j**) except the model was trained using data from replicates 1 and 2 and evaluated using data from replicate 3. The three columns in panels (**g-k**) indicate data corresponding to the three mutagenesis library blocks (block 1, left; block 2, middle; block 3, right).

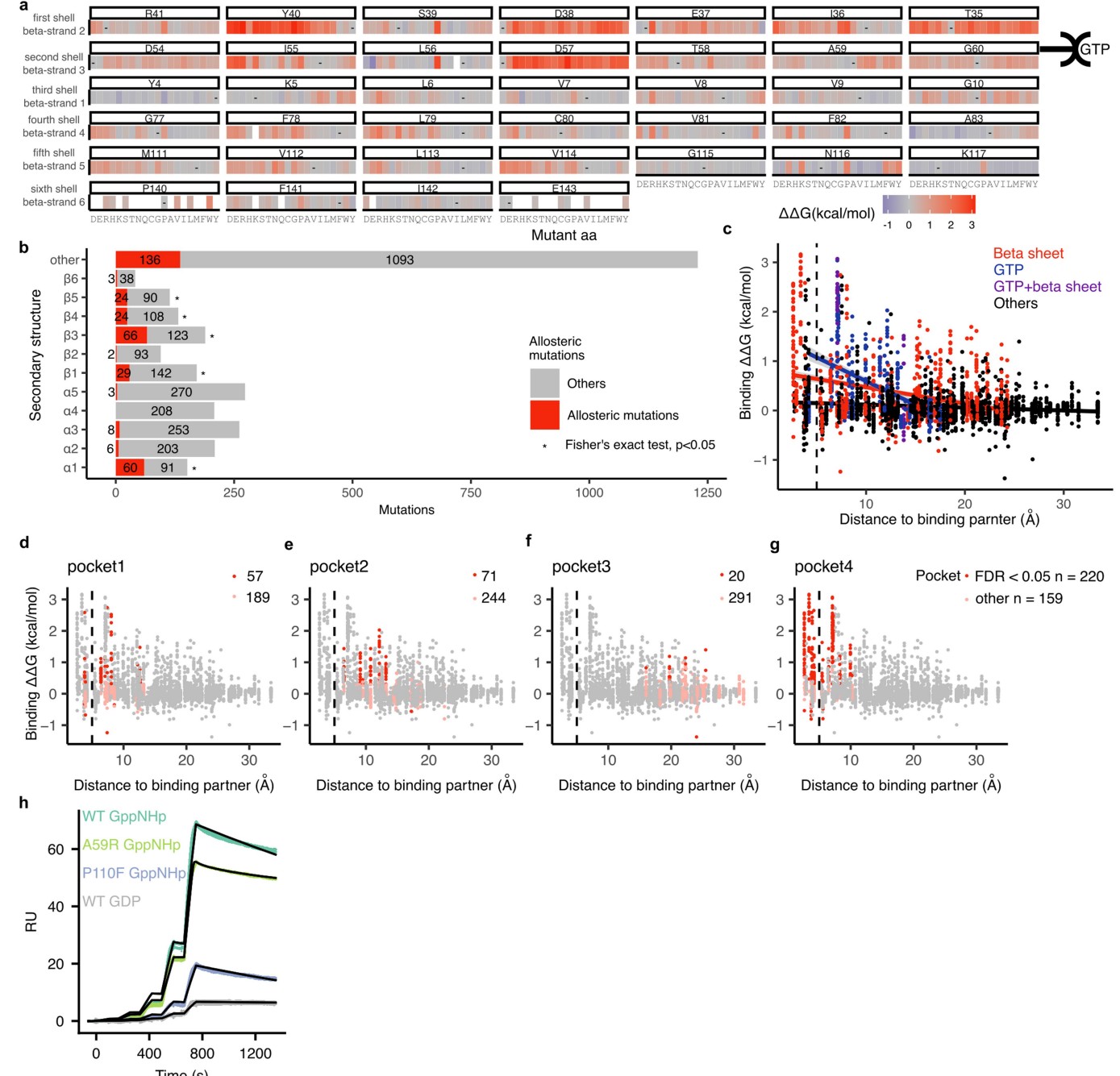

**Extended Data Fig. 2 | Allosteric mutations in the KRAS beta sheet and surface pockets. a**, Heat maps of binding free energy changes of residues in the beta sheet. GTP indicates the location of GTP in the 3D structure. **b**, Number of allosteric mutations in each secondary structure element. *, odds ratio >1, and two-sided Fisher's Exact Test P < 0.05. **c**, Scatter plot showing the binding free energy changes of all mutations and the distance to the binding partner. Residues in beta sheet and GTP binding sites (minimal side chain heavy atom distance to GTP < 5 Å) are coloured as indicated. **d**, **e**, **f**, **g**, Scatter plot showing the binding free energy changes of all mutations and the distance to the binding partner. Residues in each pocket are coloured as indicated. **h**, Surface plasmon resonance (SPR) single-cycle kinetics (SCK) measurement of the in vitro binding of a KRAS major allosteric site variant, A59R, and a pocket 3 KRAS variant P110F, both in the active state (GppNHp-loaded), to the RAF1 RBD (raw data as colour scatter plot, 2 replicates per measurement, data fit as solid black line). Analogous measurements for WT KRAS in the active (GppNHp-loaded) and inactive (GDP-loaded) states are shown for reference.

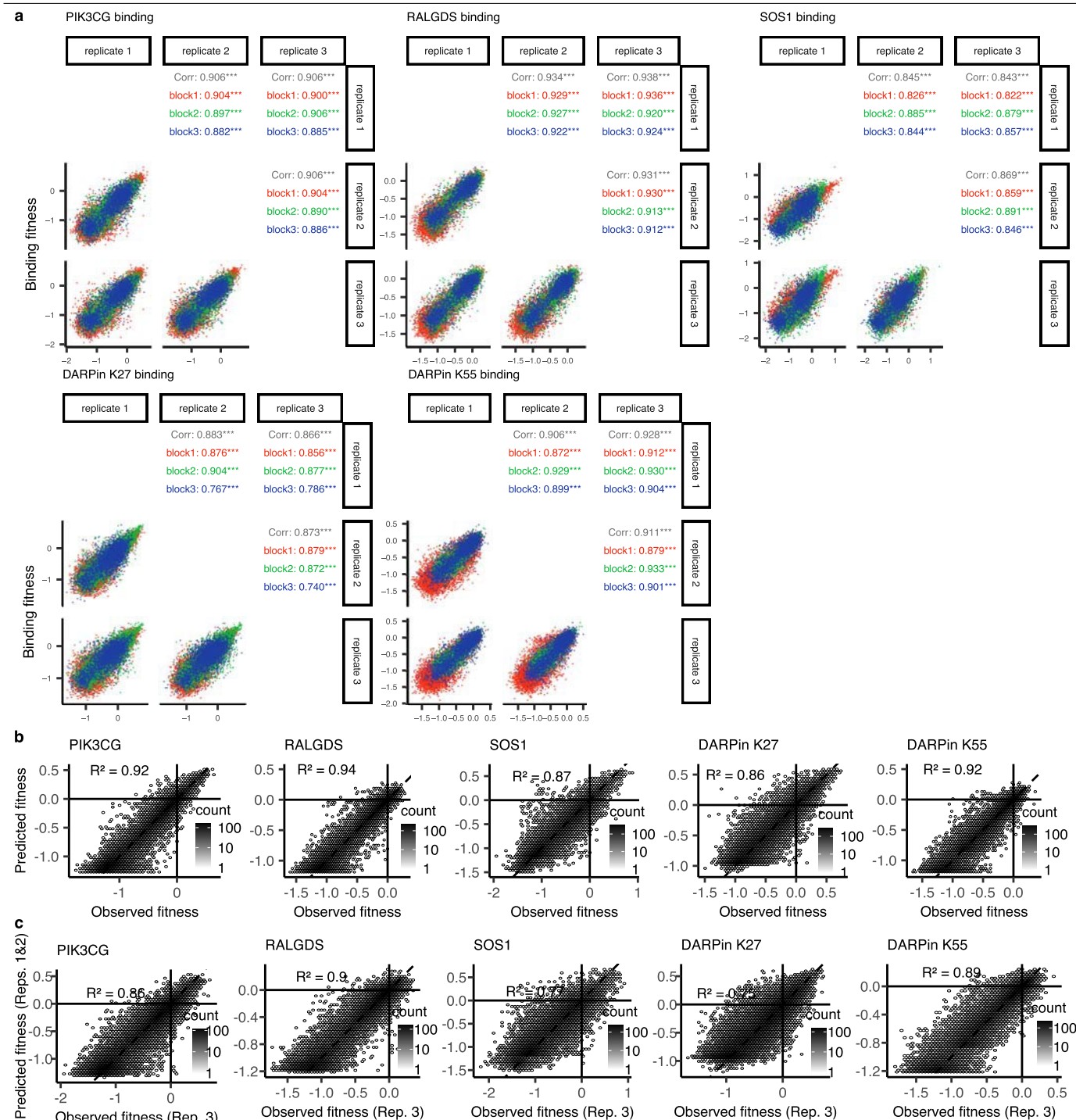

**Extended Data Fig. 3 | Experimental reproducibility and thermodynamic model fitting for five additional interaction partners. a**, Scatter plots showing the reproducibility of each block's BindingPCA fitness estimates from ddPCA. Pearson's *r* indicated on the top right corner. **b**, Performance of models fit to ddPCA data. 2D density plots comparing model predictions and observations of BindingPCA fitness for held out test data (comprising 10% of double aa substitution variants held out during model training). **c**, Same as (**b**) except the model was trained using data from replicates 1 and 2 and evaluated using data from replicate 3.

**Extended Data Fig. 4 | Seven KRAS free energy landscapes. a**, Heat maps of folding and binding free energy changes. **b**, ROC curves for predicting binding interface residues (distance to binding partner <5 Å) using weighted mean absolute binding free energy changes ($\Delta\Delta G$) in red or using weighted mean absolute binding fitness in black. AUC = Area Under the Curve.

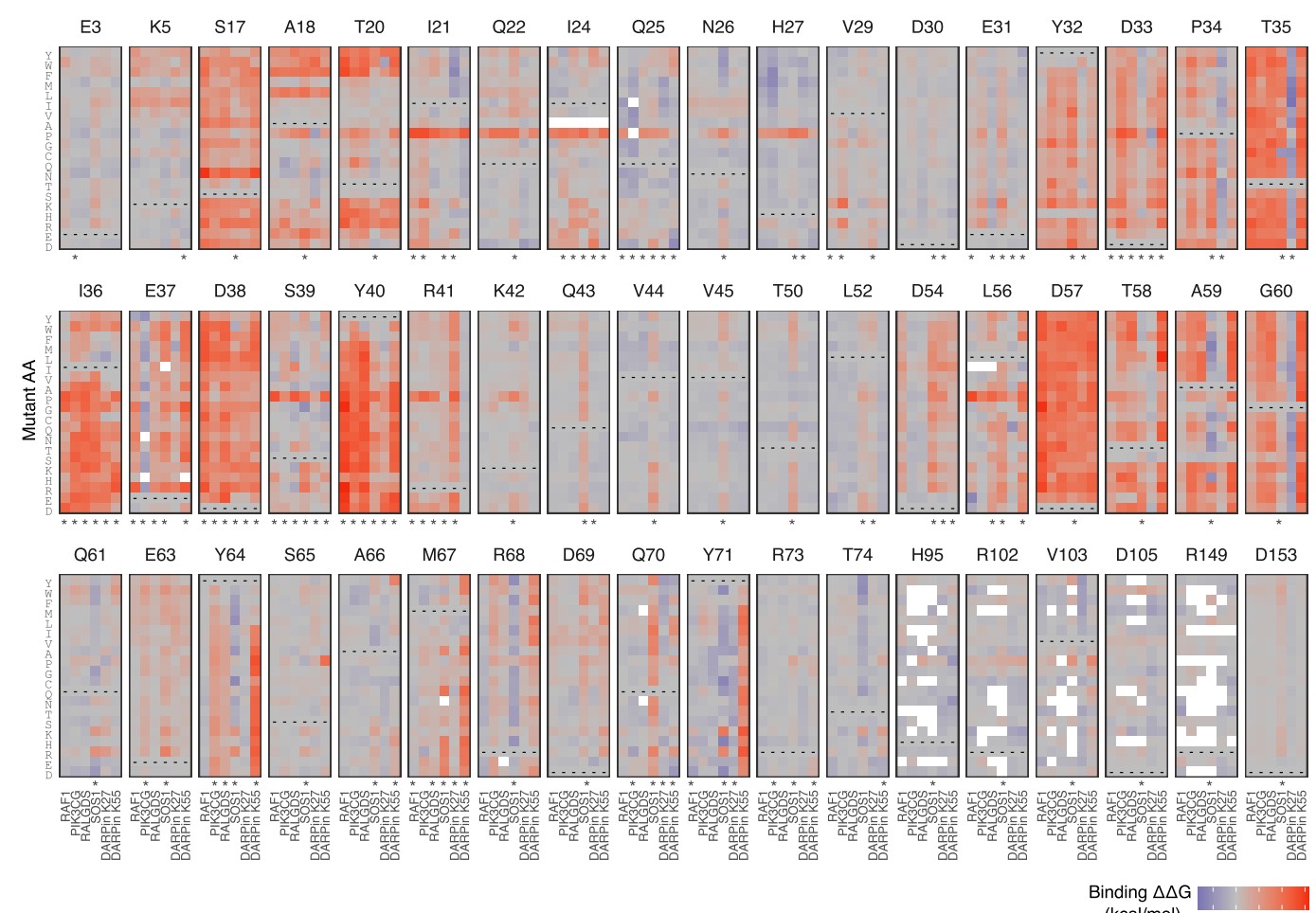

**Extended Data Fig. 5 | Binding interface specificity for all interactions.** Heat maps of binding free energy changes of all binding partners (RAF1, PIK3CG, RALGDS, SOS1, DARPin K27, DARPin K55) for binding interface residues. * indicates the binding interface residues of each binding partner.

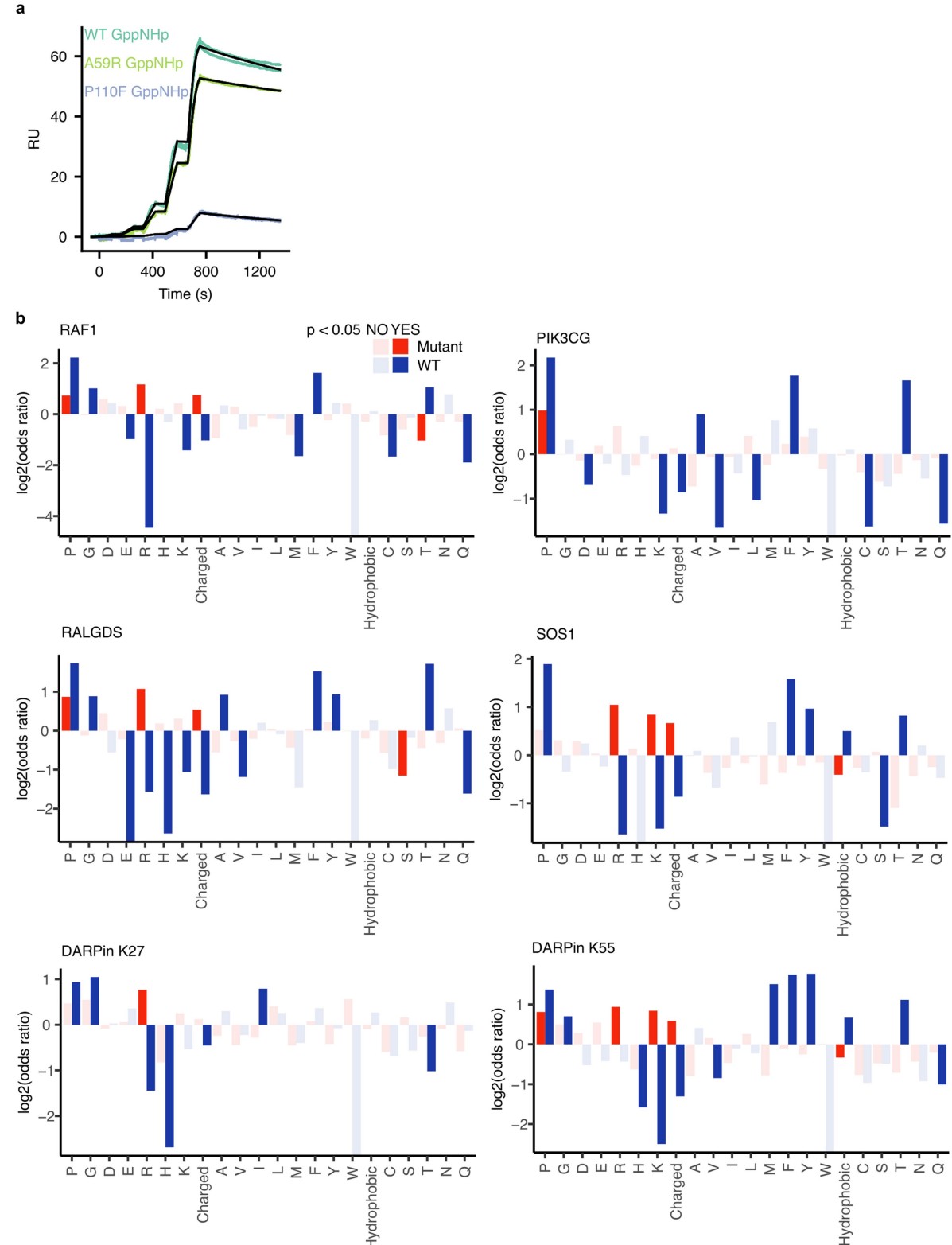

**Extended Data Fig. 6 | Allosteric regulation for six binding partners.**
**a**, Surface plasmon resonance (SPR) single-cycle kinetics (SCK) measurements of the in vitro binding of a KRAS major allosteric site variant, A59R, and a pocket 3 KRAS variant P110F, both in the active state (GppNHp-loaded) to the DARPin K55 (raw data as colour scatter plot, 2 replicates per measurement, data fit as solid black line). An analogous measurement for WT KRAS in the active state

(GppNHp-loaded) is shown for reference. **b**, Enrichments are quantified for changes from each WT aa and for changes to each aa. Enrichments are also quantified for changes from and to amino acids with particular physicochemical properties: hydrophobic (A, V, I, L, M, F, Y, W) and charged (R, H, K, D, E). Results are shown for all residues outside the binding interface.

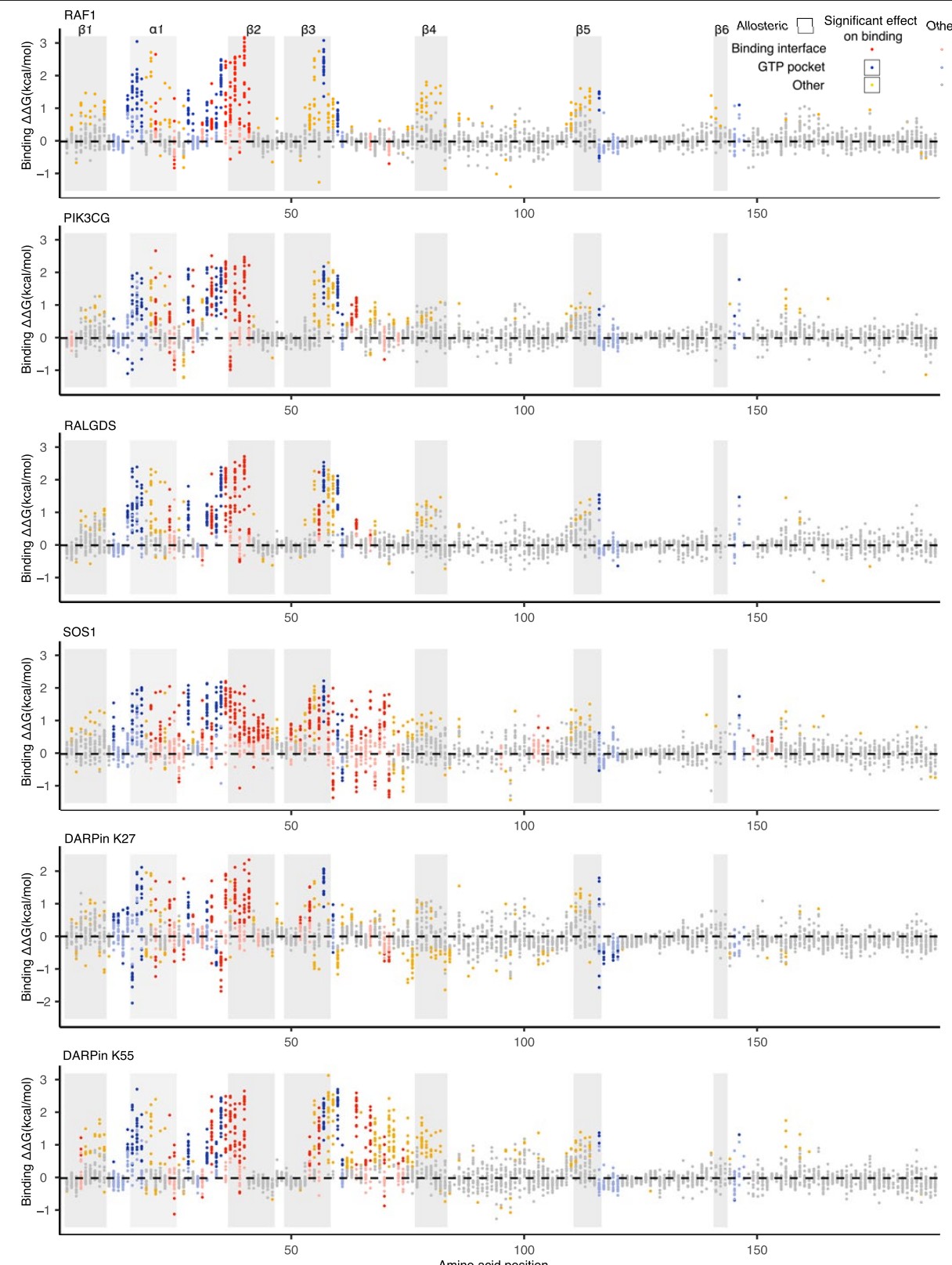

**Extended Data Fig. 7 | Binding energy and allosteric landscapes for all six binding partners.** Manhattan plots showing the binding free energy changes of all mutations coloured according to residue position and whether the free energy change is larger than the weighted mean of binding free energy changes in the binding-interfaces of all six proteins. Two-sided z test, FDR < 0.05.

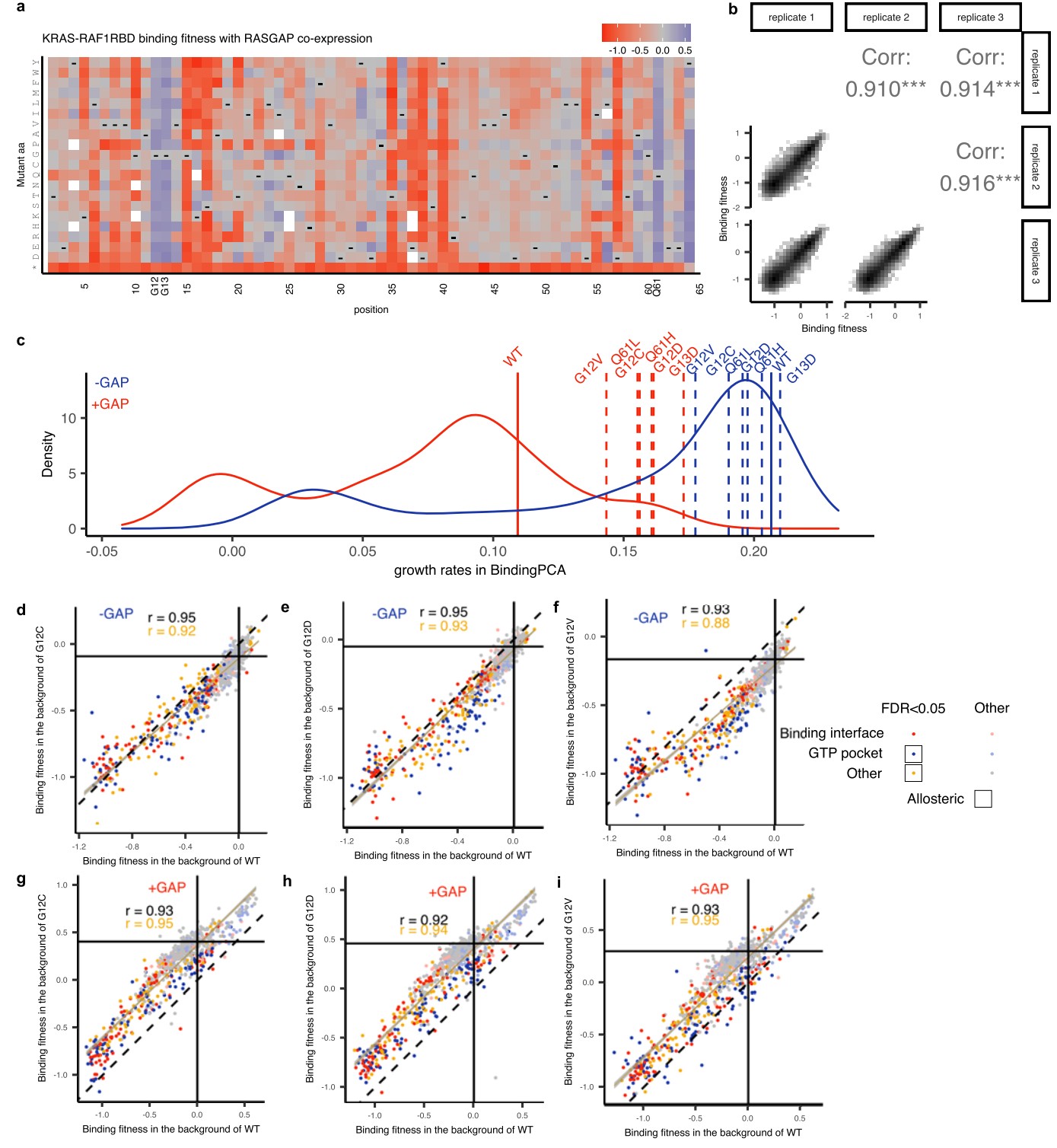

**Extended Data Fig. 8 | KRAS-RAF1RBD BindingPCA with RASGAP co-expression. a**, Heat maps of KRAS-RAF1 BindingPCA for single aa substitutions with RASGAP co-expression. White, missing values; -, WT aa; *, STOP codon. **b**, 2D density plots showing the reproducibility of binding fitness of block1 estimates from bindingPCA with RASGAP co-expression. Pearson's *r* indicated on the top right corner. **c**, Single mutation growth rates density distributions. Cancer driver mutation (G12C, G12D, G12V, G13D, Q61H, Q61L) growth rates are indicated with dashed lines. WT growth rates are indicated with solid lines. KRAS-RAF1RBD BindingPCA growth rates are coloured by blue, co-expression RASGAP KRAS-RAF1RBD BindingPCA growth rates are coloured

by red. **d-i**, Comparisons of binding fitness of single aa substitutions in WT background or G12C (left), G12D (middle), G12V (right) background for KRAS-RAF1RBD BindingPCA in the absence (top) and presence (bottom) of human GAP co-expression. Points are coloured as in Fig. 3a according to residue position and whether the corresponding binding ΔΔG is significantly greater than the weighted mean absolute binding ΔΔG of RAF1RBD BindingPCA of all mutations in the RAF1 binding interface (two-sided *z*-test, FDR = 0.05). Pearson's *r* for all mutations are indicated in black, allosteric mutations are indicated in orange.

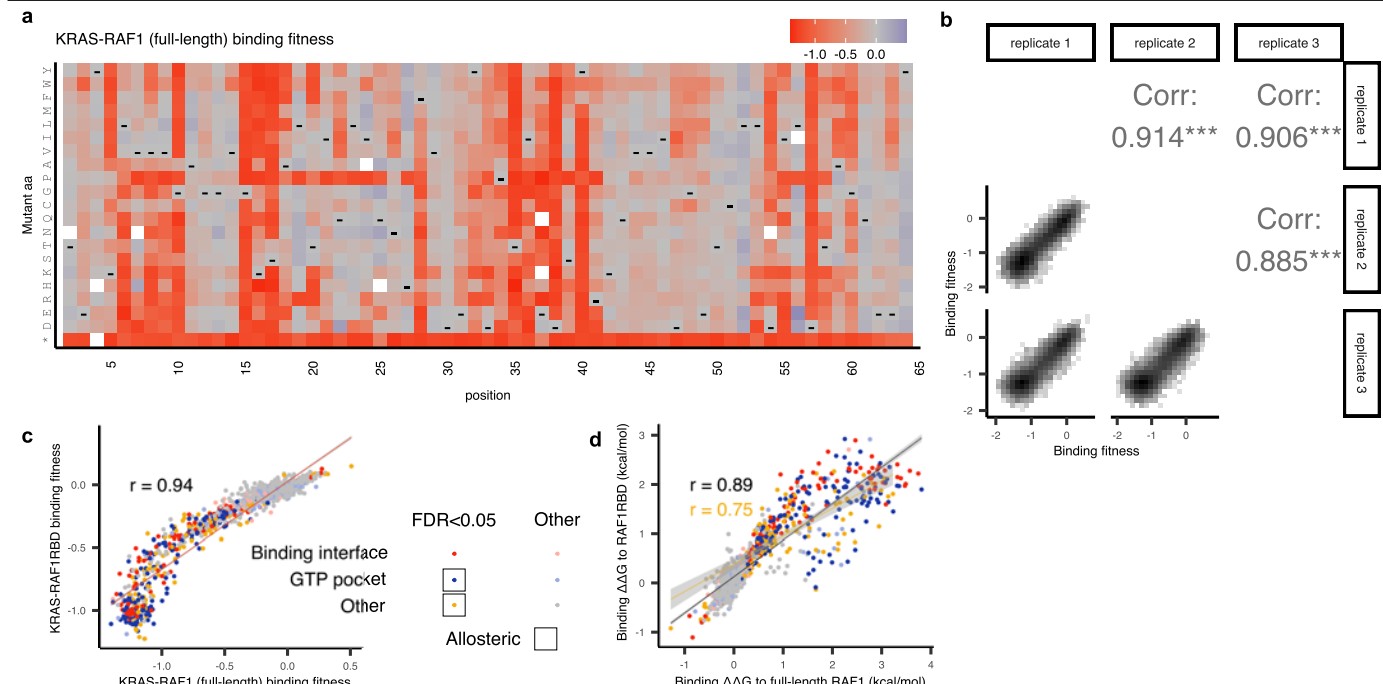

**Extended Data Fig. 9 | KRAS-full RAF1 BindingPCA. a**, Heat maps of fitness effects of single aa substitutions for KRAS-full RAF1 from BindingPCA. White, missing values; -, WT aa; *, STOP codon. **b**, 2D density plots showing the reproducibility of binding fitness of block1 for KRAS-full RAF1 from BindingPCA. Pearson's r indicated on the top right corner. **c**, Comparisons of binding fitness of single aa substitutions for KRAS-RAF1RBD and for KRAS-full RAF1 from BindingPCA. Points are coloured as in Fig. 3a according to residue position and whether the corresponding binding ΔΔG is significantly greater than the weighted mean absolute binding ΔΔG of RAF1RBD BindingPCA of all mutations in the RAF1 binding interface (two-sided z-test, FDR = 0.05). Pearson's r is shown. **d**, Comparisons of model-inferred binding free energy changes of single aa substitutions for KRAS-RAF1RBD and for KRAS-full RAF1. Points are coloured as in (**c**). Pearson's r for all mutations is indicated in black, allosteric mutations is indicated in orange.

# Reporting Summary

## Statistics

For all statistical analyses, confirm that the following items are present in the figure legend, table legend, main text, or Methods section.

| n/a | Confirmed | |
|---|---|---|
| ☐ | ☒ | The exact sample size (*n*) for each experimental group/condition, given as a discrete number and unit of measurement |
| ☐ | ☒ | A statement on whether measurements were taken from distinct samples or whether the same sample was measured repeatedly |
| ☐ | ☒ | The statistical test(s) used AND whether they are one- or two-sided *Only common tests should be described solely by name; describe more complex techniques in the Methods section.* |
| ☒ | ☐ | A description of all covariates tested |
| ☐ | ☒ | A description of any assumptions or corrections, such as tests of normality and adjustment for multiple comparisons |
| ☐ | ☒ | A full description of the statistical parameters including central tendency (e.g. means) or other basic estimates (e.g. regression coefficient) AND variation (e.g. standard deviation) or associated estimates of uncertainty (e.g. confidence intervals) |
| ☐ | ☒ | For null hypothesis testing, the test statistic (e.g. *F*, *t*, *r*) with confidence intervals, effect sizes, degrees of freedom and *P* value noted *Give P values as exact values whenever suitable.* |
| ☒ | ☐ | For Bayesian analysis, information on the choice of priors and Markov chain Monte Carlo settings |
| ☒ | ☐ | For hierarchical and complex designs, identification of the appropriate level for tests and full reporting of outcomes |
| ☐ | ☒ | Estimates of effect sizes (e.g. Cohen's *d*, Pearson's *r*), indicating how they were calculated |

*Our web collection on statistics for biologists contains articles on many of the points above.*

## Software and code

Policy information about availability of computer code

| Data collection | FastQ files from paired-end sequencing of all BindingPCA and AbundancePCA experiments were processed with DiMSum v1.2.9 using default settings with minor adjustments: https://github.com/lehner-lab/DiMSum. Experimental design files and command-line options required for running DiMSum on these datasets are available on GitHub (https://github.com/lehner-lab/krasddpcams). |
|---|---|
| Data analysis | Source code for fitting thermodynamic models (MoCHI) is available at https://github.com/lehner-lab/MoCHI. Source code for all downstream analyses and to reproduce all figures described here is available at https://github.com/lehner-lab/krasddpcams. |

For manuscripts utilizing custom algorithms or software that are central to the research but not yet described in published literature, software must be made available to editors and reviewers. We strongly encourage code deposition in a community repository (e.g. GitHub). See the Nature Portfolio guidelines for submitting code & software for further information.

# Data

Policy information about availability of data

All manuscripts must include a data availability statement. This statement should provide the following information, where applicable:

- Accession codes, unique identifiers, or web links for publicly available datasets
- A description of any restrictions on data availability
- For clinical datasets or third party data, please ensure that the statement adheres to our policy

All DNA sequencing data have been deposited in the Sequence Read Archive (SRA) under BioProject PRJNA907205: https://www.ncbi.nlm.nih.gov/bioproject/PRJNA907205

# Human research participants

Policy information about studies involving human research participants and Sex and Gender in Research.

| | |
|---|---|
| Reporting on sex and gender | *Use the terms sex (biological attribute) and gender (shaped by social and cultural circumstances) carefully in order to avoid confusing both terms. Indicate if findings apply to only one sex or gender; describe whether sex and gender were considered in study design whether sex and/or gender was determined based on self-reporting or assigned and methods used. Provide in the source data disaggregated sex and gender data where this information has been collected, and consent has been obtained for sharing of individual-level data; provide overall numbers in this Reporting Summary. Please state if this information has not been collected. Report sex- and gender-based analyses where performed, justify reasons for lack of sex- and gender-based analysis.* |
| Population characteristics | *Describe the covariate-relevant population characteristics of the human research participants (e.g. age, genotypic information, past and current diagnosis and treatment categories). If you filled out the behavioural & social sciences study design questions and have nothing to add here, write "See above."* |
| Recruitment | *Describe how participants were recruited. Outline any potential self-selection bias or other biases that may be present and how these are likely to impact results.* |
| Ethics oversight | *Identify the organization(s) that approved the study protocol.* |

Note that full information on the approval of the study protocol must also be provided in the manuscript.

# Field-specific reporting

Please select the one below that is the best fit for your research. If you are not sure, read the appropriate sections before making your selection.

☒ Life sciences          ☐ Behavioural & social sciences          ☐ Ecological, evolutionary & environmental sciences

For a reference copy of the document with all sections, see nature.com/documents/nr-reporting-summary-flat.pdf

# Life sciences study design

All studies must disclose on these points even when the disclosure is negative.

| | |
|---|---|
| Sample size | Sample sizes during the construction of the mutant libraries and the yeast competition experiments were always several fold larger than the complexity of the mutations combinations to ensure loosing as least as possible amino acid variants during the experiments. The minimum number of yeast transformants in each of the bulk competition replicate (the strongest bottleneck in the experimental design) was calculated so that the same least abundant mutations in the library (double amino acid substitutions) would be found on average in 20-25 different cells. |
| Data exclusions | Sequencing reads that did not pass the QC filters using DiMSum v1.2.9 (https://github.com/lehner-lab/DiMSum) were excluded. For all samples where the background amino acid mutations were known, variants were further filtered using a custom script to retain double AA variants consisting of single AA mutations in these designed backgrounds. |
| Replication | All bulk yeast competitions per assay and protein library were performed in triplicates. All attempts of replication were successful. |
| Randomization | Not relevant for this study. |
| Blinding | Not relevant for this study. |

# Reporting for specific materials, systems and methods

We require information from authors about some types of materials, experimental systems and methods used in many studies. Here, indicate whether each material, system or method listed is relevant to your study. If you are not sure if a list item applies to your research, read the appropriate section before selecting a response.

## Materials & experimental systems

| n/a | Involved in the study |
|-----|----------------------|
| ☒ | ☐ Antibodies |
| ☐ | ☒ Eukaryotic cell lines |
| ☒ | ☐ Palaeontology and archaeology |
| ☒ | ☐ Animals and other organisms |
| ☒ | ☐ Clinical data |
| ☒ | ☐ Dual use research of concern |

## Methods

| n/a | Involved in the study |
|-----|----------------------|
| ☒ | ☐ ChIP-seq |
| ☒ | ☐ Flow cytometry |
| ☒ | ☐ MRI-based neuroimaging |

# Eukaryotic cell lines

Policy information about cell lines and Sex and Gender in Research

| | |
|---|---|
| Cell line source(s) | Saccharomyces cerevisiae BY4742 (MATα his3Δ1 leu2Δ0 lys2Δ0 ura3Δ0) |
| Authentication | The cell line was not authenticated |
| Mycoplasma contamination | Not tested for Mycoplasma (not applicable) |
| Commonly misidentified lines (See ICLAC register) | - |

