## [Peer Review File · Nature]

Manuscript Title: The energetic and allosteric landscape for KRAS inhibition

Reviewer Comments & Author Rebuttals

Reviewer Reports on the Initial Version:

Reviewers' comments:

Referee #1:

Weng and coworkers report on a platform for high-throughput biophysics of K-Ras to characterize amino acids which influence RAF binding or protein stability. This highly quantitative assay relies on comprehensive mutagenesis covering the entire coding sequence of K-Ras which provides for highly consistent internal comparisons to measure effect sizes that might be missed by less high through put means. The authors use the dataset to search for allosteric sites within KRas that effect RAS/RAF binding. The quantitative nature of their data is exemplified by the decreasing effect size on beta strands 2-3-1-4-5-6 as each successful strand is assayed moving away from the Raf binding interaction. This gives confidence that the assays and analysis are representing what intuitively would make sense for the direct binding surface and then extending away from the binding surface. The aspect of the study that is the most novel is the analysis of the surface of Ras with respect to allosteric control of Raf binding. The two known pockets (Switch I/II and Switch II) are indeed identified by the analysis. Pocket 3 seems to be less well described and do not appear to be the subject of specific ligand screens currently. Pocket 4 seems to be part of the actual RAF binding site so it has been considered before just based on the proximity to the known effector. The high medical importance of developing additional K-Ras drugs beyond the current sotorasib/adagrasib agents makes this high throughput biophysical search of importance.

The questions which are left in my mind are:

- 1) Why weren't G12 or Q61 mutants scored in the assay? The initial basis of the different background mutants used for the initial screen are not well enough described.
- 2) Almost no consideration is given to GTP vs. GDP binding effect the mutants. The other high-throughput assays of RAS, by the Kuriyan group for example in 2017 and 2022, make detailed studies of the impact of mutations on intrinsic hydrolysis/weakened nucleotide binding--e.g. exchange mutants and GAP effects--but these critical topics are not mentioned. Is there a yeast GAP for K-RAS for example?

Referee #2:

Weng and colleagues describe their efforts to model the effect of mutations on the ability of KRAS to interact with its effectors as well as their effect on KRAS folding. They follow a systematic and unbiased approach by utilizing deep scanning mutagenesis coupled with a cellular (yeast) based sensor. Deep learning was used to infer the effect of mutations on key thermodynamic parameters. The author report novel allosteric networks that regulate effector binding and identify previously unexplored pockets which may be suitable for potential drug discovery.

Overall the study is interesting both in scope and conclusions. Its novelty is somewhat shadowed by recent saturation mutagenesis screens exploring RAS switch interactions (eLife, 2017), as well as the authors' previous work using the same approach to model SH3-mediated adaptor interactions. Nevertheless, an in-depth study of allosteric regulation of KRAS could have sufficient merit to be considered for publication in Nature. That being said, however, the current

manuscript has key limitations, especially with regards to the experimental approach and validation that must be addressed.

1. The authors indicate that the saturation mutagenesis library consists of single and double KRAS mutants. It is not clear if, and how, the later were used in the study. It is important to better describe the library and explain if and how the patterns of binding are effected in the setting of WT backbone (i.e. singly-mutated library) vs mutant backbone (i.e. doubly-mutated library).

2. Abundance PCA measures KRAS protein levels and then makes inference on folding. Other possible regulators of protein abundance seem to be ignored. A mutation could affect KRAS abundance by modulating its degradation for example. Alternatively signaling flux through effector pathways may feedback regulate KRAS abundance. It's not clear if/how these are controlled for in the experimental design, which makes the thermodynamic correlations difficult to take at face value without independent experimental validation.

3. Reliance on CRAF RBD is another limitation when studying allosteric effects. It could be that residues outside the RBD-binding pocket do in fact interact with full-length CRAF and therefore their effect may not be allosteric. The authors should have instead relied on full-length CRAF or at least CRD-RBD CRAF, since the CRD region also contacts KRAS. It is also possible that endogenous CRAF influences the exogenous Binding PCA system, either by forming intermediate complexes with KRAS (either in monomer or dimer form) or by competing with RBD for binding. In this regard, can the authors comment on the effect of residues reported to modulate KRAS dimerization on the RBD binding from their screen? A similar limitation applies to other effectors, where again sub-domains are used to infer allosteric effects. In their validation studies, the authors should use KRAS interaction-deficient RBD mutants to confirm that the measured proliferative changes are indeed specific to binding.

4. When correlating binding, free energy, and allosteric effects, the KRAS nucleotide cycle must be considered. KRAS undergoes exchange and hydrolysis and thus assumes distinct conformations in cells, so a mutation could impede CRAF binding without having orthosteric or allosteric effects on the RBD interface. For example, although oncogenic mutants at G12 or G13 enhance the interaction with RBD, they do so because they have impaired hydrolysis and not because they allosterically modulate affinity. Of course, it could also be that both are at play, but the current ms does not consider/control for this. Thus, what are currently reported as allosteric residues may in fact be residues involved in modulating exchange or hydrolysis and, as a consequence, the interaction with CRAF.

5. The key allosteric residues identified in the study need to be experimentally validated to determine that they affect binding and to independently establish the thermodynamic parameters through established biophysical methodology such as ITC or SPR. Such validation should include at a minimum both residues/mutants in the new allosteric pocket and those with effector-specific interaction. If the authors can validate their key findings and provide complementary thermodynamic findings through biophysical assays it would greatly strengthen the robustness of their conclusions.

6. I find it surprising that Fig. 1e has predominately loss-of-function mutations with few gain-of-function variants. Is this due to scaling or is the Binding PCA biased for loss of function given its dependence on MTX/DHF/THF?

7. Descriptive terms like 'central beta sheet' are confusing. The authors should use residue numbers to define the sheet. Similar comment for the reference to the allosteric pockets.

8. The methods and assays used should be described much more clearly. How long was selection in yeast for? Would the time needed to select for library expressing populations affect the screen results?

9. The authors should also experimentally demonstrate that effector-specific mutants do indeed also induce 'signaling bias' or remove this comment from their discussion. If the authors can provide experimental evidence of signaling bias this would add significant strength to the ms though.

10. It seems somewhat speculative to utilize one portion of the data to train the network and then to validate it by using the other portion of the same dataset. More robust approaches are needed to make such an assessment. (Perhaps at least using distinct training and validating datasets.)

Referee #3:

Summary of key results: The Lehner team harnessed their double deep PCA mutational scanning method (developed in Faure et al., Nature, 2022) to take a deep dive into the biophysics of the KRAS protein to identify allosteric aa positions. They train a neural network (MoCHI developed in Faure et al., Biorxiv, 2021 / Faure et al., Nature, 2022) using the mutational data and show that it accurately predicts binding $\Delta\Delta G$ and folding $\Delta\Delta G$ s. As they did for the previous study, they identify amino acid positions away from the KRAS/RAF1 binding interface and GTP binding pocket that have strong binding $\Delta\Delta G$ changes that suggest these are sites of allostery. Their allosteric network validates that all four surface pockets of the protein are allosterically active and links them to the GTP binding pocket and then to the KRAS/RAF1 binding interface mainly through the main B-sheet. They then extend their methods to six more binding partners of KRAS and show overlapping and interactor-specific ppi interfaces along with overlapping and unique ppi-specific allosteric sites. (Bonus cool finding: binding experiments of the six interactors reveal mutations that will alter specificity. Would be useful to further dissect KRAS signaling in different cancer types.) The authors suggest that because many of the allosteric sites are independent of effector, inhibiting them would negatively affect all downstream signaling.

Originality and significance: They have now shown that their double deep PCA mutational scanning method + MoCHI are able to identify allosteric sites in one whole protein (herein) and two protein domains (Faure et al., Nature, 2022), provided an exhaustive study of allosteric sites in KRAS, an important drug target, and added 22K thermodynamic measurements to the field. Although the idea that allosteric sites on proteins could be potential drug targets is not new (Deng and references therein), the Lehner lab continues to show us that high-throughput biophysics is possible and now has advanced enough experimentally and computationally to identify these sites using their extensible platform which renders the allosteric drug target idea more feasible.

Data & methodology: Raw sequence data has been deposited in SRA. It would be great if could variants/counts/scores would be made available in MAVEdb upon publication. Use of statistics is appropriate.

Conclusions: The conclusions were robust.

Suggested improvements: What would you predict the outcome of the same experiments would be if performed in the background of the G12C mutation and G12C + sotorasib? Would such an experiment begin to validate that distant allosteric are good drug candidate sites and they behave as expected in ddPCA experiments? Does such a study already exist in the literature?

"Strikingly" in last paragraph of p. 5: Are strong, non-GTP-pocket allosteric mutations enriched in B-sheet over loops or helices? A "strikingly" requires an OR or fold enrichment. Can you add a sentence as to why this is an unexpected property of the central B-sheet?

References: Fine, but RAS is not particularly my field.

Clarity and context: Text is well written and compelling. Figures are clear with intuitive data visualizations.

Author Rebuttals to Initial Comments:

Response to the reviewers:

We thank the reviewers for their enthusiasm and very constructive comments and suggestions. Please see below for point-by-point responses to each comment.

Referees' comments:

Referee #1:

Weng and coworkers report on a platform for high-throughput biophysics of K-Ras to characterize amino acids which influence RAF binding or protein stability. This highly quantitative assay relies on comprehensive mutagenesis covering the entire coding sequence of K-Ras which provides for highly consistent internal comparisons to measure effect sizes that might be missed by less high through put means. The authors use the dataset to search for allosteric sites within KRas that effect RAS/RAF binding. The quantitative nature of their data is exemplified by the decreasing effect size on beta strands 2-3-1-4-5-6 as each successful strand is assayed moving away from the Raf binding interaction. This gives confidence that the assays and analysis are representing what intuitively would make sense for the direct binding surface and then extending away from the binding surface. The aspect of the study that is the most novel is the analysis of the surface of Ras with respect to allosteric control of Raf binding. The two known pockets (Switch I/II and Switch II) are indeed identified by the analysis. Pocket 3 seems to be less well described and do not appear to be the subject of specific ligand screens currently. Pocket 4 seems to be part of the actual RAF binding site so it has been considered before just based on the proximity to the known effector. The high medical importance of developing additional K-Ras drugs beyond the current sotorasib/adagrasib agents makes this high throughput biophysical search of importance.

We thank the referee for their enthusiasm and excellent suggestions. The

questions which are left in my mind are:

- 1) Why weren't G12 or Q61 mutants scored in the assay? The initial basis of the different background mutants used for the initial screen are not well enough described.

The selection assay used in this manuscript has a large dynamic range for identifying inhibitory mutations that reduce binding of KRAS to interaction partners. It is not designed to quantify activating mutations (binding is close to saturated), even though these are, of course, also very interesting. In the revised manuscript we have performed an additional deep mutagenesis selection where we co-expressed the catalytic domain of the human GAP RASA1 (RAS p21 protein activator 1) with KRAS and the RAF1-RBD. Co-expression of the GAP very strongly reduces binding of KRAS to RAF1 in yeast (Extended Data Fig. 8c). We performed triplicate selections

for the first block of the KRAS mutagenesis library, because it contains all three of the residues most frequently mutated in cancer - G12, G13 and Q61. As shown in Extended Data Fig. 8a, mutations at G12, G13 and Q61 very strongly activate KRAS when the GAP is co-expressed in yeast.

With respect to the choice of the different background mutants used in the assay, we picked these from a pilot single mutant mutagenesis of KRAS where we selected for mutations with reduced binding to RAF1. The background mutations provide a range of starting free energies (stabilities and/or affinities) which allows us to disambiguate the folding and binding free energy changes caused by the mutations in which they are paired, as described in our previous publication describing the methods development (Faure, Domingo, Schmiedel et al. Nature 2022). Each of the blocks uses a different set of background mutations because in our sequencing strategy the background mutations need to be contained within the block being sequenced. We have now better explained this in the results section of the manuscript.

- 2) Almost no consideration is given to GTP vs. GDP binding effect the mutants. The other high-throughput assays of RAS, by the Kuriyan group for example in 2017 and 2022, make detailed studies of the impact of mutations on intrinsic hydrolysis/weakened nucleotide binding--e.g. exchange mutants and GAP effects--but these critical topics are not mentioned. Is there a yeast GAP for K-RAS for example?

Our selections and modelling are deliberately designed to quantify reduced KRAS binding affinity and/or stability, irrespective of the precise molecular mechanism. However we of course agree that the molecular mechanisms are indeed interesting. In the revised manuscript we have included additional data where we repeated RAF1 binding selections co-expressing human RASA1 (RAS p21 protein activator 1). GAP co-expression in yeast strongly inhibits KRAS binding to RAF1, consistent with it promoting GTP hydrolysis and the formation of inactive GDP-bound KRAS states. As stated above, this allows the identification of activating mutations at positions G12,G13 and Q61, in addition to inhibitory mutations.

Strikingly, these new selections show that the effects of inhibitory mutations are very conserved in the presence of human GAP co-expression and activating cancer driver mutations (G12C, G12D and G12V, $r > 0.9$, $n > 700$ mutations Extended Data Figure 8d-i). The inhibitory sites are thus effective in both wild-type KRAS and in KRAS activated by oncogenic mutations that interfere with GAP activity. This further validates the relevance of our data and the relevance of the allosteric sites that we have identified for inhibiting oncogenically active KRAS.

Referee #2:

Weng and colleagues describe their efforts to model the effect of mutations on the ability of KRAS to interact with its effectors as well as their effect on KRAS folding. They follow a systematic and unbiased approach by utilizing deep scanning mutagenesis coupled with a cellular (yeast) based sensor. Deep learning was used to infer the effect of mutations on key thermodynamic parameters. The authors report novel allosteric networks that regulate effector binding and identify previously unexplored pockets which may be suitable for potential drug discovery.

Overall the study is interesting both in scope and conclusions. Its novelty is somewhat shadowed by recent saturation mutagenesis screens exploring RAS switch interactions (eLife, 2017), as well as the authors' previous work using the same approach to model SH3-mediated adaptor interactions. Nevertheless, an in-depth study of allosteric regulation of KRAS could have sufficient merit to be considered for publication in Nature. That being said, however, the current manuscript has key limitations, especially with regards to the experimental approach and validation that must be addressed.

We thank the referee for their enthusiasm and excellent suggestions.

1. The authors indicate that the saturation mutagenesis library consists of single and double KRAS mutants. It is not clear if, and how, the latter were used in the study. It is important to better describe the library and explain if and how the patterns of binding are affected in the setting of WT backbone (i.e. singly-mutated library) vs mutant backbone (i.e. doubly-mutated library).

The double mutants are essential for disambiguating the measured changes in binding and abundance upon mutation into causal biophysical changes in the free energies of protein folding and binding (Faure, Domingo, Schmiedel et al. Nature 2022). We strongly agree that including more explanation in the current manuscript is helpful and have revised the text in the results section accordingly. There are two key experimental principles of the ddPCA approach. First, the effects of mutations on two or more molecular phenotypes—here binding and abundance—are quantified. Second, mutational effects are quantified starting from multiple genetic backgrounds. Both of these strategies are important for correctly inferring (disentangling) the underlying causal free energy changes from the measured mutational effects. Many different free energy changes can potentially underlie an observed change in phenotype (binding). We can resolve these biophysical ambiguities by quantifying how mutations interact in double mutants as well as by quantifying their effects on two different molecular traits.

We have added the following text to the results section to better re-introduce the approach:

“Protein folding and binding relate to changes in the free energies of folding (ΔG_f) and binding (ΔG_b) by nonlinear functions derived from the Boltzmann distribution (Fig. 1b)^{12,18}. Typically, many different combinations of folding and binding energy changes

could underlie a measured change in binding due to a mutation. ddPCA is an efficient experimental design to generate sufficient data to infer en masse the causal biophysical effects of mutations. There are three key principles of the approach. First, mutational effects are quantified for multiple phenotypes - here binding of KRAS to RAF1 and the abundance of KRAS in the absence of RAF1. Second, mutational effects are not only quantified in wild-type (WT) proteins but also in genetic backgrounds with altered folding and/or binding energies – here our libraries contain a median of seven double mutants for each single aa substitution in KRAS. Third, the data are used to fit a thermodynamic model in which free energy changes due to mutations combine additively in energy space (but not additively for the measured molecular phenotypes, see Methods).

We biased the choice of genetic backgrounds in our KRAS library to mutations with weak detrimental effects and used MoCHI, a substantially improved flexible package to fit user-defined mechanistic models to deep mutational scanning data using neural networks, to fit a three-state (unfolded, folded and bound KRAS) thermodynamic model to the data (see Methods) (Fig. 1c, Extended Data Fig. 1f-k). The fitted model predicts the double aa mutant data held out during training extremely well (abundance median Pearson's $R^2 = 0.74$, binding median $R^2 = 0.91$, Extended Data Fig. 1f,g,i,j) and the free energy changes are highly correlated with in vitro measurements (Pearson's $r = 0.95$, Fig. 1k)."

2. Abundance PCA measures KRAS protein levels and then makes inference on folding. Other possible regulators of protein abundance seem to be ignored. A mutation could affect KRAS abundance by modulating its degradation for example. Alternatively signaling flux through effector pathways may feedback regulate KRAS abundance. It's not clear if/how these are controlled for in the experimental design, which makes the thermodynamic correlations difficult to take at face value without independent experimental validation.

We agree that in theory a diversity of mechanisms could account for how mutations cause changes in KRAS abundance. However, in practice, the abundance measurements are well fitted ($R^2 = 0.74$) by a very simple model in which mutations only affect folding energy and folding energies combine additively in double mutants. This provides very strong evidence that the vast majority of mutations alter abundance by affecting the folding energy. Moreover, for the identification of allosteric effects, the mechanism by which mutations affect abundance is not actually important. Rather, what is important is that the model accurately predicts these abundance changes, allowing us to identify and quantify mutations that cause changes in binding that are not accounted for by changes in abundance.

3. Reliance on CRAF RBD is another limitation when studying allosteric effects. It could be that residues outside the RBD-binding pocket do in fact interact with full-length CRAF and therefore their effect may not be allosteric. The authors should have instead relied on full-length CRAF or at least CRD-RBD CRAF, since the CRD region also contacts KRAS. It is also possible that endogenous CRAF influences the exogenous Binding PCA system, either by forming intermediate complexes with KRAS (either in monomer or dimer form) or by competing with RBD for binding. In this

regard, can the authors comment on the effect of residues reported to modulate KRAS dimerization on the RBD binding from their screen? A similar limitation applies to other effectors, where again sub-domains are used to infer allosteric effects. In their validation studies, the authors should use KRAS interaction-deficient RBD mutants to confirm that the measured proliferative changes are indeed specific to binding.

This is an interesting point. In the revised manuscript we have included data from an additional triplicate set of selections where we quantified the binding of the KRAS block 1 variant library to full-length RAF1. The binding of KRAS mutants to full-length RAF1 are extremely well correlated to the binding to the RAF1-RBD (Pearson's $r = 0.94$, $n = 1,186$ genotypes, Extended Data Fig. 9c) as are the inferred binding energy changes (Pearson's $r = 0.89$, $n = 1,195$, Extended Data Fig. 9d). The allosteric mutations within block 1 also cause highly correlated changes in binding to full-length RAF1 and the RAF1-RBD (Pearson's $r = 0.75$, $n = 123$, Extended Data Fig. 9d)

4. When correlating binding, free energy, and allosteric effects, the KRAS nucleotide cycle must be considered. KRAS undergoes exchange and hydrolysis and thus assumes distinct conformations in cells, so a mutation could impede CRAF binding without having orthosteric or allosteric effects on the RBD interface. For example, although oncogenic mutants at G12 or G13 enhance the interaction with RBD, they do so because they have impaired hydrolysis and not because they allosterically modulate affinity. Of course, it could also be that both are at play, but the current ms does not consider/control for this. Thus, what are currently reported as allosteric residues may in fact be residues involved in modulating exchange or hydrolysis and, as a consequence, the interaction with CRAF.

We agree but think this is a mechanistic classification not an allosteric vs. non-allosteric classification. If a mutation is located outside of the RAF1 binding interface and it reduces the affinity of binding to KRAS, then the mutation must be allosteric. There are indeed many possible mechanisms that could mediate these allosteric effects, including alterations in GTP hydrolysis. However, this is still allosteric regulation of the protein-protein interaction. Our selections and modelling are deliberately designed to quantify reduced KRAS binding affinity and to identify allosteric mutations, irrespective of the precise molecular mechanism.

In the revised manuscript we have included additional data where we repeated RAF1 binding selections co-expressing human RASA1 (RAS p21 protein activator 1). GAP co-expression in yeast strongly inhibits KRAS binding to RAF1, consistent with it promoting GTP hydrolysis and the formation of inactive state GDP-bound KRAS states. As stated above, this allows the identification of activating mutations at positions G12, G13 and Q61, in addition to inhibitory mutations. Strikingly the effects of inhibitory mutations and allosteric mutations are very well correlated in the presence and absence of human GAP co-expression and driver mutations (Extended Data Fig. 8d-i). Thus, even with strongly perturbed nucleotide cycles due to driver mutations, the sites are still inhibitory.

5. The key allosteric residues identified in the study need to be experimentally validated to determine that they affect binding and to independently establish the thermodynamic parameters through established biophysical methodology such as ITC or SPR. Such validation should include at a minimum both residues/mutants in the new allosteric pocket and those with effector-specific interaction. If the authors can validate their key findings and provide complementary thermodynamic findings through biophysical assays it would greatly strengthen the robustness of their conclusions.

In the revised manuscript we have used surface plasmon resonance (SPR) to confirm the effects of 2 novel allosteric mutations: P110F, an allosteric mutation in pocket 3 and A59R, an allosteric mutation in a novel major allosteric site located in beta strand 3. Both mutations strongly reduced the binding of recombinant purified GTP-loaded KRAS to RAF1 in vitro (Extended Data Figure 2h) and also binding to DARPin K55 (Extended Data Fig. 6a).

6. I find it surprising that Fig. 1e has predominately loss-of-function mutations with few gain-of-function variants. Is this due to scaling or is the Binding PCA biased for loss of function given its dependence on MTX/DHF/THF?

The selection assay indeed deliberately has a large dynamic range for identifying and quantifying inhibitory mutations that reduce binding of KRAS to interaction partners. It is not designed to quantify activating gain-of-function mutations (binding is close to saturated). In the revised manuscript we have performed an additional deep mutagenesis selection where we co-expressed the catalytic domain of human RASA1 (RAS p21 protein activator 1) with KRAS and the RAF1-RBD. Co-expression of the GAP very strongly reduces binding of KRAS to RAF1 in yeast (Extended Data Fig. 8c). We performed triplicate selections for the first block of the KRAS mutagenesis library, because it contains the residues most frequently mutated in cancer - G12, G13 and Q61. As shown in Extended Data Fig. 8a, c, mutations at G12, G13 and Q61 very strongly activate KRAS when the GAP is co-expressed in yeast. Strikingly, the effects of inhibitory mutations are very well correlated in the presence and absence of human GAP co-expression, suggesting that the majority of their effects are independent of oncogenic mutation-driven perturbations in the cycling between active (GTP-bound) and inactive (GDP-bound) states (Extended Data Fig. 8d-i). In Extended Data Fig. 8d-i we have compared the effects of all single mutants in the presence and absence of G12C, G12D, G12V driver mutations (in both cases with GAP co-expression). The mutational effects are very well correlated (Pearson's $r > 0.9$, Extended Data Fig. 8d-i), showing that the inhibitory mutations, including the allosteric mutations, also inhibit oncogenically-activated KRAS.

7. Descriptive terms like 'central beta sheet' are confusing. The authors should use residue numbers to define the sheet. Similar comment for the reference to the allosteric pockets.

We have added residue numbers to define beta-strands and pockets in the figure legend. We have also reworded this sentence in the main text to read:

“Strikingly, 5 of the 8 novel major allosteric residues are located in the central (and only) six-stranded beta sheet of KRAS (Fig. 3b, c, OR = 5.24, Fisher’s exact test, $P = 2.8e-2$).”

8. The methods and assays used should be described much more clearly. How long was selection in yeast for? Would the time needed to select for library expressing populations affect the screen results?

We have added the selection times (in generations and hours) to the methods sections. These selection times are based on previous experiments and represent a trade-off between quantifying the growth rates of moderate and strongly detrimental mutations (Faure, Domingo, Schmiedel et al. Nature 2022).

9. The authors should also experimentally demonstrate that effector-specific mutants do indeed also induce ‘signaling bias’ or remove this comment from their discussion. If the authors can provide experimental evidence of signaling bias this would add significant strength to the ms though.

We have modified the text in the discussion to read ‘suggesting the potential for..’.

10. It seems somewhat speculative to utilize one portion of the data to train the network and then to validate it by using the other portion of the same dataset. More robust approaches are needed to make such an assessment. (Perhaps at least using distinct training and validating datasets.)

Withholding a portion of the same dataset during model training is standard practice in the machine learning field. As explained in the Methods section (“Thermodynamic model fitting with MoCHI”) a random 30% of double aa substitution variants was held out during training, with 20% representing the validation data and 10% representing the test data (to assess model performance).

As a further demonstration of the robustness of our method, we repeated model fitting where data from one entire selection experiment replicate was withheld from the training dataset (in addition to the held out double aa variants as explained above). In other words, fitness estimates for training were derived from two (rather than three) independent replicate selections for AbundancePCA and BindingPCA. Model performance was therefore assessed using predictions on held out double aa variants (unseen during training) and comparisons to observations in the held out independent third replicate selection (see ED Fig. 1h,k and ED Fig. 3c).

We have added this last sentence to the Results section “From molecular phenotypes to free energy changes”:

“Evaluating model performance on a held out test replicate gave similar results (abundance median $R^2 = 0.59$, binding median $R^2 = 0.88$, Extended Data Fig. 1h,k)”

Referee #3:

Summary of key results: The Lehner team harnessed their double deep PCA mutational scanning method (developed in Faure et al., Nature, 2022) to take a deep dive into the biophysics of the KRAS protein to identify allosteric aa positions. They train a neural network (MoCHI developed in Faure et al., Biorxiv, 2021 / Faure et al., Nature, 2022) using the mutational data and show that it accurately predicts binding $\Delta\Delta G$ and folding $\Delta\Delta G$ s. As they did for the previous study, they identify amino acid positions away from the KRAS/RAF1 binding interface and GTP binding pocket that have strong binding $\Delta\Delta G$ changes that suggest these are sites of allostery. Their allosteric network validates that all four surface pockets of the protein are allosterically active and links them to the GTP binding pocket and then to the KRAS/RAF1 binding interface mainly through the main B-sheet. They then extend their methods to six more binding partners of KRAS and show overlapping and interactor-specific ppi interfaces along with overlapping and unique ppi-specific allosteric sites. (Bonus cool finding: binding experiments of the six interactors reveal mutations that will alter specificity. Would be useful to further dissect KRAS signaling in different cancer types.) The authors suggest that because many of the allosteric sites are independent of effector, inhibiting them would negatively affect all downstream signaling.

Originality and significance: They have now shown that their double deep PCA mutational scanning method + MoCHI are able to identify allosteric sites in one whole protein (herein) and two protein domains (Faure et al., Nature, 2022), provided an exhaustive study of allosteric sites in KRAS, an important drug target, and added 22K thermodynamic measurements to the field. Although the idea that allosteric sites on proteins could be potential drug targets is not new (Deng and references therein), the Lehner lab continues to show us that high-throughput biophysics is possible and now has advanced enough experimentally and computationally to identify these sites using their extensible platform which renders the allosteric drug target idea more feasible.

We thank the referee for their enthusiasm and excellent suggestions.

Data & methodology: Raw sequence data has been deposited in SRA. It would be great if could variants/counts/scores would be made available in MAVEdb upon publication. Use of statistics is appropriate.

The data has now been deposited in MAVEdb (MAVEdb accession: urn:mavedb:00000115 ).

Conclusions: The conclusions were robust.

Suggested improvements: What would you predict the outcome of the same experiments would be if performed in the background of the G12C mutation and G12C + sotorasib? Would such an experiment begin to validate the that distant allosteric are good drug candidate sites and they behave as expected in ddPCA experiments? Does such a study already exist in the literature?

Thank you for the suggestion. In the revised manuscript we have performed an additional deep mutagenesis selection where we co-expressed the catalytic domain of

nature portfolio

human RASA1 (RAS p21 protein activator 1) with KRAS and the RAF1-RBD. Co-expression of the GAP very strongly reduces binding of KRAS to RAF1 in yeast (Extended Data Fig. 8c). We performed triplicate selections for the first block of the KRAS mutagenesis library, because it contains all three of the residues most frequently mutated in cancer - G12, G13 and Q61. As shown in Extended Data Fig. 8a, c, mutations at G12, G13 and Q61 very strongly activate KRAS when the GAP is co-expressed in yeast. In Extended Data Fig. 8d, g, we have compared the effects of all single mutants in the presence and absence of G12C driver mutation (in both cases with GAP co-expression). The mutational effects are very well correlated in the absence of GAP (Pearson's $r = 0.95$, $n = 843$, Extended Data Fig. 8d) as well as in the presence of GAP (Pearson's $r = 0.93$, $n = 776$, Extended Data Fig. 8g), showing that the inhibitory mutations, including the allosteric mutations, also inhibit oncogenically-activated KRAS.

"Strikingly" in last paragraph of p. 5: Are strong, non-GTP-pocket allosteric mutations enriched in B-sheet over loops or helices? A "strikingly" requires an OR or fold enrichment. Can you add a sentence as to why this is an unexpected property of the central B-sheet?

Thank you for pointing this out. We have added the odd ratio (OR = 5.24, $P = 2.8e-2$, Fisher's exact test).

References: Fine, but RAS is not particularly my field.

Clarity and context: Text is well written and compelling. Figures are clear with intuitive data visualizations.

Reviewer Reports on the First Revision:

Referees' comments:

Referee #1:

The authors have addressed my main criticism about the lack of insight into the known oncogenic mutants by performing the screen with a GAP coexpressed, and the data now included in Ext. Data Fig. 8 nicely discriminate these mutant effects when the GAP is highly expressed. It's also nice that the most active mutant in this case is the G13D mutant, which is an exchange mutant, known not to be effected by the GAP. Very nice study.

Referee #2:

The authors have addressed most of my comments in their revised manuscript. The addition of experiments with full-length CRAF (and its correlation with the RBD data) and the direct binding assessment via SPR add value and strengthen the conclusions of the manuscript. The experiments in the presence of p120 GAP also add to the study, as does the clarification that the authors have provided throughout the text. I continue to think that this is important work and potentially impactful in the field.

However, I am not convinced that some of the conclusions/interpretation of the study adhere to

the data presented. In particular, I continue to have reservations regarding the interpretation of the abundance measurements and what the authors define as allostery. These need to be addressed and clarified.

On abundance: I can appreciate the authors' rebuttal that the "mechanism by which mutations affect abundance is not actually important". I would agree with this reasoning, if the abundance measurements were used ONLY as a control for the effect of mutations on function. Abundance in the manuscript, however, is tied to folding and consequently to the calculation of free energy. This inference would be inaccurate if abundance reflects protein degradation/turnover rather than folding. Surely the authors will accept that even appropriately folded proteins can be subject to turnover. The use of abundance measurements as a proxy for folding/free energy relies on assuming that abundance is only affected by folding, which cannot be taken for granted. Furthermore, a R2 value of 0.74 (~0.5 in repeat exp) is not as high as the authors contend when stating: "a very simple model in which mutations only affect folding energy and folding energies combine additively in double mutants". The R2 values argue that the model is not as simple as assumed.

On allosteric effects:

a) The authors should provide alternative explanations for their results. It's important to highlight the difference between an allosteric effect from an indirect effect. Their reasoning in the rebuttal (which is implied in the manuscript) suggests that there is no difference between the two terms. The authors also reason: "If a mutation is located outside of the RAF1 binding interface and it reduces the affinity of binding to KRAS, then the mutation must be allosteric. There are indeed many possible mechanisms that could mediate these allosteric effects, including alterations in GTP hydrolysis". I think that either evidence or at least a discussion of mechanistic aspects that explain the allosteric effect is critical. This is particularly true for residues with known implications on the functions of RAS.

b) Since the study focuses on RAS-RAF interaction, the assumption is that any mutations in residues involved in the RAS-RAF binding interface would be orthosteric and residues outside of this interface would be allosteric. When interpreting their data the authors rely on the RAS-RBD structure, which was an understandable (albeit not ideal) starting point since the vast majority of historical structures only use the RBD domain of RAF. However, recent studies report other regions in RAF that are also involved in the interaction with RAS. The limitations of relying on such partial structures should be discussed to ensure robustness in the interpretation of the study.

Referee #3:

I am satisfied by the revision. Thank you for doing the experiment in Ext. Data Fig. 8.

Referee #4:

I was tasked with assessing the model fitting question raised by other referees, so this feedback will be specific to this point.

The authors introduced ddPCA study to KRAS to identify underlying allosteric effects through deep mutational scan data. I think this is a very good paper. The other referees also offered very good feedback. As I went over the manuscript, I had the same questions as Referee 1: Question 1 and Referee 2: Questions 2, 5, 6, and I think the authors responded mostly to my satisfaction so I won't repeat them here. The new experiments done in revision addressed these points quite well.

In terms of the data processing, Referee 2 raised the question about potentially using a different

train/validation/test split. While the authors were correct that the random hold out of the same dataset is the common practice, there are some caveats specific to this data:

Main question: Often a random held out test data assumes the holdout set is independent to minimize model overfit. In this case, the authors held out 30% random data (20% to validation and 10% to test) from the double mutant set, and I think generally this is acceptable. However, I do not see why the authors only chose to hold out from the double mutant data and not from all data. If anything, this is making the tasks too easy for the model, I think, and could not avoid overfitting. The authors' response with holding out full replica data does not fully address this issue, since replica data by definition should be redundant. One would expect fitting quality to redundant data be very high.

That said, neural networks almost always overfit, and I do believe even if overfit, it should not change the conclusion of the paper. However, if the authors can elaborate on the rationale for holding out only double mutant data, it would offer better guidance to the fitted ddG. It is very nice to see the high correlation between inferred ddG and experimental values shown in Figs. 1k and 4b, but it could be more stringent if all or some of these mutations with experimental binding ddG could be assigned to the test dataset (and with all instances removed from training). This way the experimental value offers even better validation.

Minor comments: Ext. Data Fig. 1 is very low resolution; the fonts are pixelated and hard to read.

Author Rebuttals to First Revision:

Referees' comments:

Referee #1:

The authors have addressed my main criticism about the lack of insight into the known oncogenic mutants by performing the screen with a GAP coexpressed, and the data now included in Ext. Data Fig. 8 nicely discriminate these mutant effects when the GAP is highly expressed. It's also nice that the most active mutant in this case is the G13D mutant, which is an exchange mutant, known not to be effected by the GAP. Very nice study.

Thank you very much.

Referee #2:

The authors have addressed most of my comments in their revised manuscript. The addition of experiments with full-length CRAF (and its correlation with the RBD data) and the direct binding assessment via SPR add value and strengthen the conclusions of the manuscript. The experiments in the presence of p120 GAP also add to the study, as does the clarification that the authors have provided throughout the text. I continue to think that this is important work and potentially impactful in the field.

However, I am not convinced that some of the conclusions/interpretation of the study adhere to the data presented. In particular, I continue to have reservations regarding the interpretation of the abundance measurements and what the authors define as allostery. These need to be addressed and clarified.

On abundance: I can appreciate the authors' rebuttal that the "mechanism by which mutations affect abundance is not actually important". I would agree with this reasoning, if the abundance measurements were used ONLY as a control for the effect of mutations on function. Abundance in the manuscript, however, is tied to folding and consequently to the calculation of free energy. This

inference would be inaccurate if abundance reflects protein degradation/turnover rather than folding. Surely the authors will accept that even appropriately folded proteins can be subject to turnover. The use of abundance measurements as a proxy for folding/free energy relies on assuming that abundance is only affected by folding, which cannot be taken for granted. Furthermore, a R2 value of 0.74 (~0.5 in repeat exp) is not as high as the authors contend when stating: "a very simple model in which mutations only affect folding energy and folding energies combine additively in double mutants". The R2 values argue that the model is not as simple as assumed.

We agree that a diversity of mechanisms could account for how mutations cause changes in KRAS abundance. However, we disagree that the reported R-squared values suggest that a more complex model would better describe the AbundancePCA data. Using DiMSum fitness errors – which report on the technical noise in the data (<https://doi.org/10.1186/s13059-020-02091-3>) – we estimate the total explainable variance in the AbundancePCA data for block1 to be <79%. Therefore, our models capture the vast majority of explainable (genetic) variance in this dataset (0.74/0.79 = 94%, ED Fig. 1g). The performance on the held out replicate 3 data is lower ($R^2 \sim 0.5$ i.e. Pearson's $r \sim 0.7$) as expected – averaging over biological replicates reduces technical noise – yet we estimate this is close to the total explainable variance in this single replicate given the inter-replicate correlations for the abundance measurements (median Pearson's $r = 0.69$ for block1, ED Fig. 1b). In summary, a simple two-state model in which mutations overwhelmingly affect folding energy and combine additively in double mutants is very strongly supported by the results.

On allosteric effects:

a) The authors should provide alternative explanations for their results. It's important to highlight the difference between an allosteric effect from an indirect effect. Their reasoning in the rebuttal (which is implied in the manuscript) suggests that there is no difference between the two terms. The authors also reason: "If a mutation is located outside of the RAF1 binding interface and it reduces the affinity of binding to KRAS, then the mutation must be allosteric. There are indeed many possible mechanisms that could mediate these allosteric effects, including alterations in GTP hydrolysis". I think that either evidence or at least a discussion of mechanistic aspects that explain the allosteric effect is critical. This is particularly true for residues with known implications on the functions of RAS.

We agree and have added the following text to the discussion:

"Finally, we note that there are likely multiple molecular mechanisms that mediate the allosteric effects, including shifts in conformational equilibria, altered nucleotide binding or hydrolysis, and propagated structural and dynamic perturbations in the binding interfaces. Future experiments will be needed to disentangle the mechanistic causes of allostery."

b) Since the study focuses on RAS-RAF interaction, the assumption is that any mutations in residues involved in the RAS-RAF binding interface would be orthosteric and residues outside of this interface would be allosteric. When interpreting their data the authors rely on the RAS-RBD structure, which was an understandable (albeit not ideal) starting point since the vast majority of historical structures only use the RBD domain of RAF. However, recent studies report other regions in RAF that are also involved in the interaction with RAS. The limitations of relying on such partial structures should be discussed to ensure robustness in the interpretation of the study.

We agree and have added the following text to the discussion:

"A second potential caveat of our experiments was that we quantified binding of KRAS to isolated RAS-binding domains (RBDs) and, in general, mutations that have allosteric effects in isolated domains may have different effects or directly participate in binding in full-length proteins."

Referee #3:

I am satisfied by the revision. Thank you for doing the experiment in Ext. Data Fig. 8.

Thank you very much.

Referee #4:

I was tasked with assessing the model fitting question raised by other referees, so this feedback will be specific to this point.

The authors introduced ddPCA study to KRAS to identify underlying allosteric effects through deep mutational scan data. I think this is a very good paper. The other referees also offered very good feedback. As I went over the manuscript, I had the same questions as Referee 1:Question1 and Referee 2:Questions 2,5,6, and I think the authors responded mostly to my satisfaction so I won't repeat them here. The new experiments done in revision addressed these points quite well.

In terms of the data processing, Referee 2 raised the question about potentially using a different train/validation/test split. While the authors were correct that the random hold out of the same dataset is the common practice, there are some caveats specific to this data:

We thank the referee for their enthusiasm and excellent suggestions.

Main question: Often a random held out test data assumes the holdout set is independent to minimize model overfit. In this case, the authors held out 30% random data (20% to validation and 10% to test) from the double mutant set, and I think generally this is acceptable. However, I do not see why the authors only chose to hold out from the double mutant data and not from all data. If anything, this is making the tasks too easy for the model, I think, and could not avoid overfitting. The authors' response with holding out full replica data does not fully address this issue, since replica data by definition should be redundant. One would expect fitting quality to redundant data be very high.

That said, neural networks almost always overfit, and I do believe even if overfit, it should not change the conclusion of the paper. However, if the authors can elaborate on the rationale for holding out only double mutant data, it would offer better guidance to the fitted ddG. It is very nice to see the high correlation between inferred ddG and experimental values shown in Figs. 1k and 4b, but it could be more stringent if all or some of these mutations with experimental binding ddG could be assigned to the test dataset (and with all instances removed from training). This way the experimental value offers even better validation.

When fitting thermodynamic models we held out a fraction of the double mutant data only in order to evaluate whether our assumption that free energy changes due to mutations combine additively in energy space was justified. The fact that the simple three-state model predicts the held out double aa mutant data so well strongly supports this assumption. We have updated this sentence in the results section "From phenotypes to free energy changes" to emphasise this point:

*"The fitted model predicts the double aa mutant data held out during training extremely well (abundance median $R^2 = 0.74$, binding median $R^2 = 0.91$, Extended Data Fig. 1f,g,i,j) **strongly supporting the assumption that most free energy changes combine additively in doubles** and these inferred free energy changes are highly correlated with in vitro measurements (Pearson's $r = 0.95$, Fig. 1k)."*

We also tested whether the two suggestions by this reviewer regarding the training data had any impact on the model performance. Namely, we repeated model fitting while both (i) entirely excluding data from single mutants present in *in vitro* (experimental) binding $\Delta\Delta G$ s used for model validation and (ii) randomly holding out both single and double mutant data from the training set (rather than double mutants only).

Correlations of inferred free energy changes in this more stringent “New model” with the independent *in vitro* validation data are extremely similar to results using the “Original model” (equivalent to Fig. 1k and Fig. 4b,c).

Likewise, direct comparisons between inferred free energies from these models show that they are extremely similar (Pearson's $r > 0.97$).

Minor comments: Ext. Data Fig. 1 is very low resolution; the fonts are pixelated and hard to read.

A higher resolution Fig is uploaded. Thank you for noticing this.

Reviewer Reports on the Second Revision:

Referees' comments:

Referee #2:

None

Referee #4:

Remarks to the Author:

The authors addressed my concerns satisfyingly. As expected, the change in data split does not change the conclusion, but the more stringent test indeed alleviates overfitting with a slightly lowered correlation. I do not see the manuscript using the "new model" results to replace the original figures. I strongly encourage the authors to make such an update since the original model indeed shows signs of overfitting, and it is better to report the more stringent model. Even though the performance is largely correlated, holding out only the double mutant data, in my opinion, really seems like an unusual practice.

Author Rebuttals to Second Revision:

Referees' comments:

Referee #2:

(This referee stated confidentially that the responses largely satisfied their requests.)

Thank you. In addition, as requested by the editor, we have added the following text to the manuscript:

“Thermodynamic model fitting with MoCHI”

“We also assume that mutation effects on abundance level predominantly arise from folding free energy changes. However, protein abundance can be influenced by factors beyond folding, such as degradation or cellular processes which may skew the free energy estimates.”

Referee #4:

The authors addressed my concerns satisfyingly. As expected, the change in data split does not change the conclusion, but the more stringent test indeed alleviates overfitting with a slightly lowered correlation. I do not see the manuscript using the "new model" results to replace the original figures. I strongly encourage the authors to make such an update since the original model indeed shows signs of overfitting, and it is better to report the more stringent model. Even though the performance is largely correlated, holding out only the double mutant data, in my opinion, really seems like an unusual practice.

1. As shown in the previous response, we repeated model fitting according to reviewer #4's suggestion and the inferred free energies are extremely similar ($r \geq 0.97$). The correlation plots are shown below:

2. Regarding the issue of overfitting, we respectfully disagree with the assertion that a higher correlation of model parameters with in vitro measurements is indicative of overfitting.

Overfitting occurs when a model performs well on training data but poorly on unseen test data. The plots below show that the original model ('Random doubles') performs better on both training and test data. There is no justification for changing to the alternative model.

3. Indeed the original model performs (slightly) better (not worse) on independent data than the alternative model. These in vitro measurements of binding energies are made using a completely different technique. This again argues for sticking to the original model.

Our decision to restrict the held-out data to double mutants was motivated by the need to explicitly test the key assumption of additivity of free energy changes in our model, a fundamental principle also emphasized in the original ddpca paper (Faure, Domingo, Schmiedel et al. Nature 2022). The justification for this choice is supported by our results.

In addition there was a second motivation: the relatively higher quality of measurements for single mutants, as evidenced by higher read counts due to the hierarchical nature of the DMS library (Faure, Schmiedel et al. Genome Biol, 2020), is an important consideration. Including singles in the test dataset corresponds to holding out almost an order of magnitude more sequencing reads during model training than when the test set is restricted to doubles. This is likely why the original model has slightly better performance. In summary, our results robustly demonstrate that our choice of variants for the test dataset does not lead to overfitting.